# Threshold-Guided Optimization for Visual Generative Models

**Jinbin Bai** [1,2]  **Yu Lei** [1]  **Qingyu Shi** [3]  **Aosong Feng** [4]  **Yi Xin** [5]  **Zhuoran Zhao** [1]  **Fei Shen** [1]  **Kaidong Yu** [1]
**Xiangtai Li** [3]

## Abstract

Aligning large visual generative models with human feedback is often performed through pairwise preference optimization. While such approaches are conceptually simple, they fundamentally rely on annotated pairs, limiting scalability in settings where feedback is collected as independent scalar ratings. In this work, we revisit the KL-regularized alignment objective and show that the optimal policy implicitly compares each sample's reward to an instance-specific baseline that is generally intractable. We propose a threshold-guided alignment framework that replaces this oracle baseline with a data-driven global threshold estimated from empirical score statistics. This formulation turns alignment into a binary decision task on unpaired data, enabling effective optimization directly from scalar feedback. We also incorporate a confidence weighting term to emphasize samples whose scores deviate strongly from the threshold, improving sample efficiency. Experiments across both diffusion and masked generative paradigms, spanning three test sets and five reward models, show that our method consistently improves preference alignment over previous methods. These results position our threshold-guided framework as a simple yet principled alternative for aligning visual generative models without paired comparisons.

## 1. Introduction

Aligning large generative models with human feedback is a central challenge during the post-training stage. Reinforcement Learning from Human Feedback (RLHF) (Christiano et al., 2017; Achiam et al., 2023; Ouyang et al., 2022; Stiennon et al., 2020) formulates this problem as the optimization

[1]National University of Singapore [2]Collov Labs [3]Peking University [4]Yale University [5]Shanghai Innovation Institute. Correspondence to: Jinbin Bai <jinbin.bai@u.nus.edu>.

*Proceedings of the 43rd International Conference on Machine Learning*, Seoul, South Korea. PMLR 306, 2026. Copyright 2026 by the author(s).

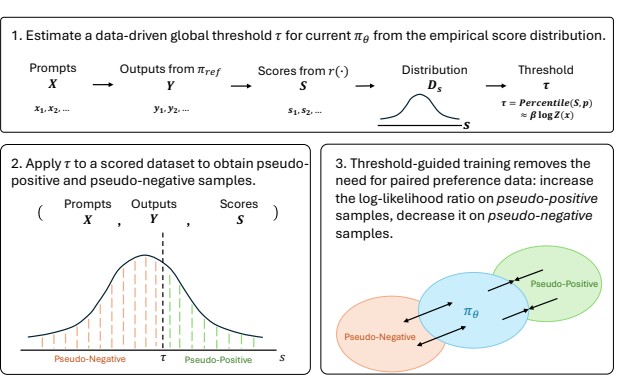

*Figure 1.* The overview of the threshold-guided optimization framework for visual generative models. Scalar feedback is converted into pseudo-positive/pseudo-negative labels via a data-driven threshold, with confidence weighting based on the distance to the threshold.

of a KL-regularized policy objective, demonstrating that complex human values can be incorporated through learned rewards. However, the operational complexity and instability of RL-style optimization (Rafailov et al., 2023) have motivated a shift towards simpler, more stable policy-fitting methods. Direct Preference Optimization (DPO) reframes alignment as a classification problem over pairs of preferred $(y_w)$ and rejected $(y_l)$ responses. Building on closed-form solutions of KL-regularized reinforcement learning objectives (Peters & Schaal, 2007; Peng et al., 2019; Go et al., 2023), DPO sidesteps explicit reward modeling and RL rollouts: the intractable partition function $Z(x)$ cancels when taking differences between two responses (Rafailov et al., 2023), yielding an objective equivalent to fitting a reparameterized Bradley-Terry model (Bradley & Terry, 1952).

A central limitation of DPO (Rafailov et al., 2023) and its successors (Meng et al., 2024; Ethayarajh et al., 2024; Liu et al., 2024) is their fundamental reliance on paired preference data. In many practical settings, especially for visual generative models, feedback is more naturally collected as unpaired samples with scalar scores: for instance, 1–5 star ratings from users or continuous outputs from a reward model. While such data can in principle be converted into pairs (*e.g.*, by comparing scores within a batch), this conversion is ad hoc, discards information about the absolute scale of scores, and can amplify noise when scores cluster

tightly. This creates a gap between the pairwise assumptions baked into DPO-style objectives and the scalar nature of many real-world feedback signals.

In this work, we propose a *threshold-guided optimization framework* that operates directly on unpaired scalar feedback, as illustrated in Figure 1. The key idea is to convert absolute scores into preference-style supervision through a **thresholding mechanism**: Samples whose scores exceed a data-driven threshold are treated as "pseudo-positive", while those below the threshold are treated as "pseudo-negative". This yields a binary decision signal on unpaired data, enabling a DPO-like classification loss without explicitly constructing pairs. We further introduce a **confidence-weighting mechanism** that scales each sample's contribution by its deviation from the threshold, allowing the model to learn more from high-confidence examples while still using the full dataset.

Our framework is grounded in a theoretical analysis of the KL-regularized alignment objective (Eq. 1). We show that for any individual sample, whether to increase or decrease its probability relative to a reference model, the optimal policy update is governed by an elegant but intractable decision rule: the sample's reward must be compared against an instance-dependent oracle baseline $\tau^*(x) = \beta \log Z(x)$. This perspective makes explicit the core obstacle that has constrained both RLHF and policy-fitting methods: the intractability of the baseline $\tau^*(x)$ for each input $x$. Our threshold-guided framework introduces a **tractable proxy** for this ideal rule by replacing the oracle baseline with a global threshold $\tau$ estimated from empirical score statistics. This reformulation turns preference alignment into a binary classification problem on unpaired data, and the resulting estimator enjoys desirable consistency and calibration properties. The confidence weighting further refines this proxy by leveraging the full magnitude of the scalar scores.

We validate the proposed framework under two dominant paradigms in vision-centric generative modeling: diffusion-based models (Ho et al., 2020) trained with mean squared error (MSE) objectives, and MaskGIT-style masked token models (Chang et al., 2022) trained with cross-entropy. Our empirical evaluation covers multiple popular foundation models (Stable Diffusion v1.5 (Rombach et al., 2022), Meissonic (Bai et al., 2024b), FLUX (Black-Forest-Labs, 2024), and Wan 1.3B (Wan et al., 2025)) and five widely used reward models (HPSv2.1 (Wu et al., 2023), PickScore (Kirstain et al., 2023), ImageReward (Xu et al., 2023), CLIP Score (Radford et al., 2021), and LAION Aesthetic Score (Schuhmann et al., 2022)). Across these settings, threshold-guided optimization consistently improves preference alignment over previous optimization methods. Together, these findings position our framework as a simple yet principled approach to preference alignment for visual generative models without relying on paired comparisons.

In summary, our contributions are as follows:

- We introduce a threshold-guided alignment framework for visual generative models that operates directly on unpaired scalar feedback, addressing a key limitation of existing pairwise preference optimization methods.

- We provide a principled derivation of this framework as a tractable approximation to the KL-optimal decision rule, revealing how a data-driven threshold and confidence weighting arise naturally from the underlying objective.

- We demonstrate through comprehensive experiments on diffusion and masked generative paradigms that threshold-guided optimization achieves consistent preference alignment gains over previous optimization methods across multiple reward models.

## 2. Related Work

**Generative Models.** Generative modeling has witnessed a rapid evolution, from Generative Adversarial Networks (GANs) (Goodfellow et al., 2020) and Variational Autoencoders (VAEs) (Kingma et al., 2013), to the current dominance of diffusion models (Sohl-Dickstein et al., 2015; Ho et al., 2020; Podell et al., 2023; Betker et al., 2023; Black-Forest-Labs, 2024; Bai et al., 2024a; 2023) and masked generative transformers (Chang et al., 2022; Bai et al., 2024b; Shi et al., 2025; Bai et al., 2025). Denoising diffusion probabilistic models (DDPMs) have established a new state-of-the-art in high-fidelity image synthesis by iteratively reversing a noise-injection process. Their remarkable generative quality and training stability have made them the *de-facto* architecture for large-scale text-to-image systems. A key breakthrough enabling granular control over the generation process was classifier-free guidance (Ho & Salimans, 2022), which allows a trade-off between sample fidelity and diversity without an external classifier. While classifier-free guidance provides a powerful mechanism for conditioning on explicit text prompts, it does not inherently address alignment with more abstract or ineffable human preferences, such as aesthetic appeal, compositional coherence, or stylistic nuance. This limitation necessitates a more direct approach to learn from human or model-based feedback.

**Preference Optimization for Language Models.** The challenge of aligning powerful base models with human intent was first studied systematically in large language models (LLMs). Reinforcement Learning from Human Feedback (RLHF) (Christiano et al., 2017) formulates alignment as optimizing a KL-regularized policy objective using a learned reward model. The standard RLHF

pipeline (Ouyang et al., 2022; Achiam et al., 2023) consists of supervised fine-tuning (SFT), reward model training on human preference labels, and reinforcement learning (typically PPO (Schulman et al., 2017)) to maximize the learned reward under a KL constraint to a reference policy. Despite its empirical success, RLHF is complex and brittle in practice, requiring multiple models and inheriting the optimization instability of deep RL.

These difficulties have motivated a family of *policy fitting* methods that optimize directly on preference data. Direct Preference Optimization (DPO) (Rafailov et al., 2023) casts alignment as binary classification over pairs of preferred and rejected responses. By exploiting a closed-form solution to the KL-regularized objective, DPO directly relates this classification loss to the optimal policy, bypassing explicit reward modeling and RL rollouts. Subsequent work has generalized the basic formulation along several axes, including improvements to variance and regularization (Meng et al., 2024; Liu et al., 2025c) and kernel- or transportation-based objectives (Ethayarajh et al., 2024).

More recently, several works have sought to relax the reliance on strictly paired preferences and to exploit more general feedback structures. From a probabilistic inference viewpoint, Abdolmaleki et al. (2024) propose a framework that can learn from unpaired positive and negative examples, derived via an expectation–maximization procedure and regularized by a KL constraint to a reference policy. In parallel, Matrenok et al. (2025) introduces Quantile Reward Policy Optimization (QRPO), which fits policies directly to scalar rewards by transforming them into quantiles. This transformation makes the induced reward distribution uniform and renders the partition function analytically tractable, leading to a simple regression-style objective while still operating within the KL-regularized policy optimization framework. Our work is conceptually related to these methods. We revisit the KL-regularized alignment objective and seek a tractable surrogate that can be optimized from scalar feedback. However, instead of quantile transformations or EM-based inference, we focus on a threshold-guided formulation that converts scores into binary decisions via a data-driven baseline, together with a confidence-weighted loss.

**Preference Optimization for Vision Models.** Following the success of preference optimization in LLMs, similar ideas have been adapted to vision and, in particular, diffusion models (Lee et al., 2023; Yang et al., 2024b; Deng et al., 2024; Yang et al., 2024a; Li et al., 2024; Ren et al., 2025; Yang et al., 2025; Zhang et al., 2024). Early work largely mirrored the RLHF pipeline, first training an explicit reward or aesthetic model from human judgments (Schuhmann et al., 2022) and then fine-tuning the diffusion process with RL, as in DDPO (Black et al., 2023). While effective, these methods inherit the complexity and instability of RL-based training.

Inspired by DPO, a line of work has proposed direct preference optimization for diffusion (Wallace et al., 2024), adapting the DPO loss to the diffusion setting to improve aesthetics and prompt faithfulness without explicit RL. Related approaches such as generalized reinforcement policy optimization (GRPO) (Xue et al., 2025; Liu et al., 2025a) further explore KL-regularized updates for generative models. However, most of these methods fundamentally rely on pairwise preference data, or on pairs synthesized from scalar scores, and therefore do not fully exploit the information contained in absolute ratings. In practice, feedback for visual generation is often available as unpaired scalar scores: either from human raters or from learned reward models.

In contrast, we focus on *threshold-guided optimization* for visual generative models under scalar feedback. We revisit the same KL-regularized objective underlying RLHF and DPO and derive a tractable surrogate based on comparing scores to a data-driven threshold. This yields a classification-style loss that can be optimized directly on unpaired, scored data, providing a complementary path to preference alignment that does not require explicit preference pairs or RL-style training. Our work is also related to recent visual preference-optimization methods that relax standard pairwise supervision. Diffusion-KTO (Li et al., 2024) operates on desirable and undesirable samples under a Kahneman-Tversky objective, while RankDPO (Karthik et al., 2025) uses ranked or listwise supervision. By contrast, TGO is designed for independently scored samples: the scalar score determines both the threshold-induced pseudo-label and the confidence weight, without requiring explicit pairs or ranked candidate groups.

## 3. Method

We begin by revisiting the KL-regularized objective that underlies modern preference-based alignment methods and highlight the role of an instance-dependent oracle baseline. We then show how pairwise policy fitting methods such as DPO exploit this structure to avoid the intractable partition function, and why this breaks down in the unpaired, scalar-feedback setting. Finally, we derive a threshold-guided surrogate objective that enables direct optimization from unpaired scores and instantiate it for visual generative models.

### 3.1. Preliminaries: KL-Regularized RL and Policy Fitting

Many alignment methods seek a policy $\pi_\theta$ that maximizes expected reward while remaining close to a reference policy $\pi_{\text{ref}}$ (Ziebart et al., 2010; Jaques et al., 2019). This is captured by the KL-regularized objective

$$\max_{\pi_\theta} \mathbb{E}_{x \sim \mathcal{D}, y \sim \pi_\theta(\cdot|x)} \big[\mathcal{R}(x,y)\big] - \beta \, \mathbb{D}_{\text{KL}}(\pi_\theta(\cdot|x) \,\|\, \pi_{\text{ref}}(\cdot|x)) \quad (1)$$

where $\mathcal{R}(x, y)$ is a reward function and $\beta$ controls the strength of regularization.

This objective admits a closed-form optimal policy

$$\pi^*(y|x) = \frac{1}{Z(x)} \pi_{\text{ref}}(y|x) \exp\left(\frac{1}{\beta} \mathcal{R}(x, y)\right), \quad (2)$$

where $Z(x) = \sum_y \pi_{\text{ref}}(y|x) \exp(\mathcal{R}(x, y)/\beta)$ is a per-input partition function. Directly optimizing through Eq. (2) is intractable because $Z(x)$ requires summing over all possible outputs $y$, an infinite space in realistic language and vision models.

Taking logarithms and rearranging yields

$$\mathcal{R}(x, y) = \beta \log \frac{\pi^*(y|x)}{\pi_{\text{ref}}(y|x)} + \beta \log Z(x). \quad (3)$$

The log-ratio $\log \frac{\pi^*(y|x)}{\pi_{\text{ref}}(y|x)}$ is thus determined by the reward shifted by an *oracle baseline* $\tau^*(x) = \beta \log Z(x)$:

$$\log \frac{\pi^*(y|x)}{\pi_{\text{ref}}(y|x)} > 0 \iff \mathcal{R}(x, y) > \tau^*(x). \quad (4)$$

In words, the KL-optimal decision rule increases the probability of a sample if and only if its reward exceeds an instance-dependent baseline $\tau^*(x)$, which is itself intractable because it depends on $Z(x)$.

**Pairwise policy fitting.** DPO (Rafailov et al., 2023) and related methods circumvent the need to compute $\tau^*(x)$ by working with *pairwise* preferences. Under the Bradley-Terry model (Bradley & Terry, 1952), the probability that $y_w$ is preferred to $y_l$ given $x$ is

$$p(y_w \succ y_l \mid x) = \sigma\big(\mathcal{R}(x, y_w) - \mathcal{R}(x, y_l)\big), \quad (5)$$

where $\sigma(\cdot)$ is the logistic function. Substituting Eq. (3) into Eq. (5) gives

$$\mathcal{R}(x, y_w) - \mathcal{R}(x, y_l) = \beta \log \frac{\pi_\theta(y_w|x)}{\pi_{\text{ref}}(y_w|x)} - \beta \log \frac{\pi_\theta(y_l|x)}{\pi_{\text{ref}}(y_l|x)}, \quad (6)$$

in which the oracle baseline $\tau^*(x)$ cancels out. This leads to the DPO loss

$$\mathcal{L}_{\text{DPO}}(\pi_\theta; \pi_{\text{ref}}) = -\mathbb{E}_{(x, y_w, y_l) \sim \mathcal{D}}\Big[\log \sigma\Big(\beta \log \frac{\pi_\theta(y_w|x)}{\pi_{\text{ref}}(y_w|x)} - \beta \log \frac{\pi_\theta(y_l|x)}{\pi_{\text{ref}}(y_l|x)}\Big)\Big], \quad (7)$$

which optimizes a classification-style objective without ever computing $Z(x)$.

Crucially, this derivation relies on access to *paired* samples $(y_w, y_l)$ for the same prompt $x$. When supervision is provided instead as *unpaired* scalar scores $s$ for individual samples, the pairwise cancellation in Eq. (6) is no longer available. In this setting, we must confront the oracle baseline $\tau^*(x)$ more directly.

## 3.2. Threshold-Guided Optimization from Scalar Feedback

We consider the common situation where supervision is available as unpaired scalar feedback:

$$\mathcal{D} = \{(x_i, y_i, s_i)\}_{i=1}^n,$$

where $s_i \in \mathbb{R}$ is a score from a human annotator or a reward model. We treat $s$ as a noisy but informative proxy for the latent reward $\mathcal{R}(x, y)$ in Eq. (1). Our goal is to design a tractable surrogate objective that (approximately) follows the KL-optimal decision rule in Eq. (4) without computing $\tau^*(x)$ or constructing explicit preference pairs.

### 3.2.1. IDEAL KL-OPTIMAL DECISION RULE

Rewriting Eq. (4) in terms of the policy ratio gives

$$\pi^*(y|x) \begin{cases} > \pi_{\text{ref}}(y|x), & \text{if } \mathcal{R}(x, y) > \tau^*(x), \\ < \pi_{\text{ref}}(y|x), & \text{if } \mathcal{R}(x, y) < \tau^*(x). \end{cases} \quad (8)$$

As shown in Appendix A, the policy ratio $\pi^*(y|x)/\pi_{\text{ref}}(y|x)$ is strictly increasing in $\mathcal{R}(x, y)$, so Eq. (8) indeed defines an optimal classification rule. The difficulty is that the decision boundary $\tau^*(x) = \beta \log Z(x)$ is input-dependent and intractable.

### 3.2.2. DATA-DRIVEN THRESHOLDING AND PSEUDO-PREFERENCES

We approximate the ideal rule in Eq. (8) using two standard modeling assumptions: First, the observed score $s$ is a monotone transform of the latent reward $\mathcal{R}(x, y)$, so comparing rewards can be approximated by comparing scores. Second, the oracle baseline $\tau^*(x)$ can be replaced by a *data-driven threshold* $\tau$ estimated from the empirical score distribution.

Concretely, given scores $\{s_i\}$, we set

$$\tau = \text{Percentile}(\{s_i\}, p),$$

with $p$ typically chosen as the median ($p=0.5$). We then define a pseudo-preference label

$$l = \mathbb{1}[s \geq \tau],$$

so that samples with $s \geq \tau$ are treated as "pseudo-positive" (desirable) and those with $s < \tau$ as "pseudo-negative" (undesirable). This yields the following surrogate decision rule:

$$(x, y, s) \mapsto \begin{cases} \pi_\theta(y|x) \gtrsim \pi_{\text{ref}}(y|x), & \text{if } s \geq \tau, \\ \pi_\theta(y|x) \lesssim \pi_{\text{ref}}(y|x), & \text{if } s < \tau, \end{cases} \quad (9)$$

which mimics the sign of the KL-optimal update using a single global threshold. In practice, $\tau$ can be estimated once per epoch or from a proxy dataset generated by the reference policy (Sec. 3.5).

## 3.3. Threshold-Guided Objective

To turn the surrogate rule in Eq. (9) into a learning objective, we define an *implicit policy score*

$$\hat{s}_{\theta,\text{ref}}(x,y) = \beta \log \frac{\pi_\theta(y|x)}{\pi_{\text{ref}}(y|x)},$$

which corresponds to the log-ratio in Eq. (3). We then train a classifier that predicts whether a sample should be above or below the threshold:

$$\mathcal{L}_{\text{TG}}(\pi_\theta) = -\mathbb{E}_{(x,y,s)\sim\mathcal{D}}\Big[w(s,\tau)\big(l \log \sigma(\hat{s}_{\theta,\text{ref}}) $$
$$+ (1-l)\log(1-\sigma(\hat{s}_{\theta,\text{ref}}))\big)\Big],$$
(10)

where $\sigma(\cdot)$ is the logistic function, $l = \mathbb{1}[s \geq \tau]$, and $w(s,\tau)$ is a confidence weight.

**Confidence weighting.** Scores far from the threshold provide less ambiguous supervision than scores near $\tau$. We encode this intuition by setting

$$w(s,\tau) = 1 + c|s-\tau|,$$

with a hyperparameter $c \geq 0$ controlling the strength of reweighting. This places more emphasis on high-confidence samples while still using the full dataset.

Minimizing $\mathcal{L}_{\text{TG}}$ guides $\pi_\theta$ to assign larger log-probability ratios to pseudo-positive samples than to pseudo-negative ones, thereby implementing a threshold-guided approximation to the KL-optimal rule in Eq. (4).

## 3.4. Theoretical Properties

Although the surrogate objective in Eq. (10) is motivated by tractability, it is important to understand its statistical behavior. Our analysis (Appendix B) studies the estimator that minimizes the *population* version of $\mathcal{L}_{\text{TG}}$ and then relates the empirical minimizer to this population optimum.

**Theorem 3.1** (Informal guarantees). *Let $\theta^\star$ denote the population minimizer of $\mathcal{L}_{\text{TG}}$ under mild regularity conditions, and let $\hat{\theta}_n$ be the empirical minimizer on $n$ i.i.d. samples. Then:*

1. **Consistency.** *$\hat{\theta}_n \to \theta^\star$ as $n \to \infty$; in particular, the empirical estimator converges to the population optimum of the surrogate objective.*

2. **Asymptotic bias.** *The estimation error admits an expansion of order $O(1/n)$ whose leading term depends on the curvature, variance, and skewness of $\mathcal{L}_{\text{TG}}$ (see Appendix B).*

3. **Calibration.** *The classifier induced by the threshold $\tau$ approximates the KL-optimal decision rule*

*$\mathbb{1}[\mathcal{R}(x,y) > \tau^*(x)]$ up to a quantifiable estimation error that vanishes as the score distribution is estimated more accurately.*

These results show that the threshold-guided surrogate is statistically well-behaved: the empirical minimizer is consistent for the population optimum, and the deviation from the ideal KL rule is controlled by the quality of the score and threshold estimates. We emphasize that the guarantees are stated with respect to the surrogate objective in Eq. (10), which is designed to approximate the original KL-regularized problem while remaining tractable.

## 3.5. Implementation for Visual Generative Models

### 3.5.1. LOG-LIKELIHOOD APPROXIMATION

The implicit policy score $\hat{s}_{\theta,\text{ref}}$ in Eq. (10) depends on log-likelihoods $\log \pi_\theta(y|x)$ and $\log \pi_{\text{ref}}(y|x)$, whose computation is model-dependent.

**Diffusion models (continuous outputs).** For diffusion-based generative models, exact likelihoods are generally intractable. Following prior work on diffusion preference optimization (Lee et al., 2023; Wallace et al., 2024), we approximate the likelihood under a Gaussian observation model. If $\hat{y}_\theta(x)$ is the denoised prediction, we write

$$p(y|x) = \mathcal{N}\big(\hat{y}_\theta(x), \sigma^2 I\big)$$
$$\Rightarrow \log p(y|x) = -\frac{1}{2\sigma^2}\big\|y - \hat{y}_\theta(x)\big\|^2 + \text{const.}$$
(11)

and thus use a scaled negative MSE as a surrogate:

$$\log \pi_\theta(y|x) \approx -\frac{1}{T}\text{MSE}\big(y, \hat{y}_\theta(x)\big),$$

with a temperature hyperparameter $T$ controlling the scale.

**MaskGIT (discrete outputs).** For MaskGIT (Chang et al., 2022)-style (masked generative transformers) models, an image $y$ is encoded as tokens $(t_1, \ldots, t_N)$ via a VQ-GAN (Esser et al., 2021). The model predicts masked tokens given visible context and condition $x$, so the log-likelihood is directly available as

$$\log \pi_\theta(y|x) = \frac{1}{|M|}\sum_{i\in M}\log p_\theta(t_i \mid y_{\backslash M}, x),$$

where $M$ is the set of masked positions and $y_{\backslash M}$ denotes unmasked tokens.

### 3.5.2. TRAINING PROCEDURE

In our experiments, we adopt an offline training setup, where a fixed dataset $\{(x_i, y_i)\}$ is first generated and scored. Reward scores $s_i = r(x_i, y_i)$ are precomputed, and a global

**Algorithm 1** Offline Threshold-Guided Optimization from Scalar Feedback

---

**Require:** Initial policy $\pi_\theta$, reward model $r(\cdot)$, dataset $\mathcal{D} = \{(x_i, y_i)\}$, batch size $B$, percentile $p$, scale $c$, temperature $\beta$
**Ensure:** Updated policy $\pi_\theta$
1: $\pi_{\text{ref}} \leftarrow \pi_\theta$
2: Compute $s_i \leftarrow r(x_i, y_i)$ for all $(x_i, y_i) \in \mathcal{D}$
3: $\tau \leftarrow \text{Percentile}(\{s_i\}, p)$
4: **for** each epoch **do**
5:   **for** each minibatch $\{(x_j, y_j, s_j)\}_{j=1}^{B} \sim \mathcal{D}$ **do**
6:     **for** $j = 1, \ldots, B$ **do**
7:       $l_j \leftarrow \mathbb{1}[s_j \geq \tau]$
8:       $w_j \leftarrow 1 + c \cdot |s_j - \tau|$
9:       $\hat{r}_j \leftarrow \beta\big(\log \pi_\theta(y_j|x_j) - \log \pi_{\text{ref}}(y_j|x_j)\big)$
10:       $\ell_j \leftarrow -w_j\Big(l_j \log \sigma(\hat{r}_j) + (1 - l_j)\log(1 - \sigma(\hat{r}_j))\Big)$
11:     **end for**
12:     $\mathcal{L} \leftarrow \frac{1}{B}\sum_{j=1}^{B} \ell_j$
13:     Update $\pi_\theta$ by a gradient step on $\nabla_\theta \mathcal{L}$
14:   **end for**
15:   (Optional) $\pi_{\text{ref}} \leftarrow \pi_\theta$
16: **end for**

---

threshold $\tau$ is estimated from their empirical distribution (or from a proxy set generated by $\pi_{\text{ref}}$). During training, each mini-batch receives pseudo-labels $l$ and confidence weights $w(s, \tau)$, and the model parameters are updated by minimizing $\mathcal{L}_{\text{TG}}$. Algorithm 1 summarizes the threshold-guided optimization procedure.

In large-scale settings where pre-scored datasets may come from a distribution different from the current policy, we estimate $\tau$ using smaller proxy sets generated by the reference policy and scored by the reward model, and then reuse the resulting threshold on the large dataset. As the proxy sets grow, the estimation error in $\tau$ shrinks, and our theoretical analysis (Theorem 3.1) shows that the induced bias in the surrogate objective decays at rate $O(1/n)$.

## 4. Experiments

We evaluate Threshold-Guided Optimization (TGO) for visual generation alignment under two training settings: (i) a true scalar-feedback setting where each prompt is associated with a single scalar score (our 10k-prompt collection), and (ii) a pairwise-to-scalar setting derived from Pick-a-Pic v2, used only for controlled comparison with prior preference-alignment methods that are commonly benchmarked on this dataset. Across both settings, we report results on standard prompt test sets and multiple reward models to reduce sensitivity to any single scorer.

### 4.1. Experimental Setup

**Training data.** (1) **Scalar-feedback prompts (10k)**. We collect and filter 10,000 high-quality text prompts from the Internet. For each foundation model, we sample images

conditioned on these prompts and obtain scalar feedback via reward models. We use a percentile-based threshold to convert scalar scores into *pseudo-preferred* vs. *pseudo-rejected* labels. (2) **Pick-a-Pic v2 (pairwise-to-scalar)**. Pick-a-Pic v2 contains human pairwise preferences. To follow the common evaluation protocol in diffusion alignment and to enable fair comparison with baselines, we convert the pairwise annotations into per-image scalar scores by aggregating win counts across comparisons[1], then apply the same percentile thresholding to obtain pseudo-preferred and pseudo-rejected subsets. We use this setting only for comparison with methods whose official benchmarks are reported on Pick-a-Pic v2 (Sec. 4.3).

**Models and training.** We fine-tune three text-to-image foundation models: Stable Diffusion (Rombach et al., 2022), Meissonic (Bai et al., 2024b), and FLUX (Black-Forest-Labs, 2024). Unless otherwise stated, TGO uses $\beta = 1$, diffusion log-likelihood temperature $T = 0.001$, and confidence scaling $c = 5$. For the scalar-feedback (10k) setting, we train TGO and SFT with identical optimization hyper-parameters (batch size 128, 78 update steps, learning rate 1e−5). For Pick-a-Pic v2 comparisons, we follow the published or official training protocol of each baseline.

**Baselines.** We compare against: (i) the original pretrained model, (ii) SFT (training on pseudo-preferred samples only), (iii) CSFT (Wu et al., 2023), and preference-alignment baselines including AlignProp (Prabhudesai et al., 2023), Diffusion-DPO (Wallace et al., 2024), Diffusion-KTO (Li et al., 2024), and DSPO (Zhu et al., 2025).

**Evaluation protocol.** We evaluate on three standard prompt test sets: Pick-a-Pic (424 prompts) (Kirstain et al., 2023), PartiPrompts (1,632 prompts) (Yu et al., 2022), and HPSv2 (3,200 prompts) (Wu et al., 2023). We report scores from multiple widely used reward models (HPSv2.1 (Wu et al., 2023), PickScore (Kirstain et al., 2023), CLIP (Radford et al., 2021), ImageReward (Xu et al., 2023) and LAION Aesthetic Score (Schuhmann et al., 2022)) to reduce dependence on any single scorer.

### 4.2. Main Results on 10k Scalar-Feedback Prompts

**Quantitative results.** We first study the true scalar-feedback setting using our 10k-prompt collection. Tab. 2 reports mean and median scores on HPS Prompts under four reward models. Compared to the original model and SFT, TGO yields consistent improvements across founda-

---

[1]We explicitly note that **Diffusion-KTO (Li et al., 2024) is also trained under an unpaired-data interface** (desirable vs. undesirable sets derived from preferences); we therefore adopt the same pairwise-to-scalar conversion to keep the comparison aligned.

*Table 1.* Mean reward-model scores of different post-training methods applied to SD v1.5 on three text-to-image benchmarks. Higher is better. The best result in each column is in **bold**, and the second best is underlined.

| Dataset | Method | PickScore | HPSv2.1 | CLIPScore | ImageReward | Aesthetic |
|---------|--------|-----------|---------|-----------|-------------|-----------|
| Pick-a-Pic | SD v1.5 | 20.35 | 0.2469 | 26.84 | 0.1131 | 5.33 |
| | SFT | 20.68 | 0.2724 | 27.04 | 0.5015 | 5.53 |
| | CSFT | 20.48 | 0.2649 | 26.83 | 0.3473 | 5.49 |
| | AlignProp | 20.35 | 0.2469 | 26.84 | 0.1131 | 5.33 |
| | Diffusion-DPO | 20.78 | 0.2594 | 27.45 | 0.3433 | 5.38 |
| | Diffusion-KTO | 20.94 | 0.2814 | 27.48 | 0.6381 | 5.51 |
| | DSPO | 20.35 | 0.2469 | 26.84 | 0.1131 | 5.33 |
| | TGO (ours) | **21.05** | **0.2860** | **27.62** | **0.6703** | **5.55** |
| PartiPrompts | SD v1.5 | 21.15 | 0.2475 | 26.59 | 0.2426 | 5.25 |
| | SFT | 21.31 | 0.2673 | 26.73 | 0.4383 | 5.48 |
| | CSFT | 21.20 | 0.2617 | 26.63 | 0.3197 | 5.45 |
| | AlignProp | 21.15 | 0.2475 | 26.59 | 0.2426 | 5.25 |
| | Diffusion-DPO | 21.41 | 0.2561 | 27.02 | 0.3547 | 5.32 |
| | Diffusion-KTO | 21.50 | 0.2757 | 27.06 | 0.6269 | 5.47 |
| | DSPO | 21.15 | 0.2475 | 26.59 | 0.2426 | 5.25 |
| | TGO (ours) | **21.55** | **0.2790** | **27.15** | **0.6574** | **5.51** |
| HPSv2 | SD v1.5 | 20.71 | 0.2455 | 28.90 | 0.1384 | 5.29 |
| | SFT | 21.09 | 0.2733 | 28.79 | 0.4744 | 5.51 |
| | CSFT | 20.89 | 0.2654 | 28.74 | 0.3703 | 5.46 |
| | AlignProp | 20.64 | 0.2412 | 28.82 | 0.1066 | 5.29 |
| | Diffusion-DPO | 21.14 | 0.2593 | 29.31 | 0.3672 | 5.39 |
| | Diffusion-KTO | 21.24 | 0.2873 | 30.02 | 0.7365 | 5.50 |
| | DSPO | 20.64 | 0.2412 | 28.82 | 0.1066 | 5.29 |
| | TGO (ours) | **21.45** | **0.2961** | **30.38** | **0.7595** | **5.53** |

*Table 2.* Quantitative comparison of text-to-image generation across original, supervised fine-tuned (SFT), and threshold-guided optimization (TGO) methods. Higher is better. Evaluation is conducted on HPS Prompts using four different reward models.

| Paradigm | Method | HPSv2.1 (↑) | | PickScore (↑) | | ImageReward (↑) | | Aesthetic (↑) | |
|----------|--------|-------------|--------|---------------|--------|-----------------|--------|---------------|--------|
| | | Mean | Median | Mean | Median | Mean | Median | Mean | Median |
| Diffusion | SD v1.4 | 0.2454 | 0.2462 | 20.8040 | 20.7784 | 0.1406 | 0.1773 | 5.4277 | 5.4293 |
| | +SFT | 0.2506 | 0.2520 | 20.7217 | 20.7006 | 0.2348 | 0.2870 | 5.4927 | 5.4948 |
| | +TGO | **0.2618** | **0.2631** | **20.9001** | **20.8907** | **0.3523** | **0.4246** | **5.6036** | **5.6122** |
| MaskGIT | Meissonic | 0.2810 | 0.2837 | 21.8315 | 21.7686 | 0.8230 | 0.9674 | 5.7692 | 5.7578 |
| | +SFT | 0.2912 | 0.2928 | 21.9105 | 21.8419 | 0.9215 | 1.0985 | 5.8013 | 5.7999 |
| | +TGO | **0.2915** | **0.2934** | **21.9421** | **21.8946** | **0.9369** | **1.1233** | **5.8270** | **5.8234** |

tion models, with gains reflected not only in mean scores but also in medians, indicating a broad distributional shift rather than improvements driven by a small subset of prompts.

**Distributional analysis.** To further quantify how TGO changes model behavior beyond averages, Fig. 3 visualizes full score distributions on SD v1.4 across reward metrics. This view complements table summaries and makes it explicit whether improvements correspond to a global right-shift or only to tail effects.

### 4.3. Comparison on Pick-a-Pic v2

While Pick-a-Pic v2 provides pairwise human preferences, most diffusion alignment baselines report results on this dataset. To compare against these methods under their standard setting, we follow prior work[2] and convert Pick-a-Pic

---

[2]**Important note on the "unpaired" protocol.** Diffusion-KTO is trained through an unpaired interface (desirable/undesirable sets) derived from preference annotations. We therefore adopt the same

v2 into per-image scalar scores via aggregated win counts, then threshold at percentile $p = 0.5$ to obtain pseudo-preferred and pseudo-rejected subsets. This yields 237,530 pseudo-preferred and 690,538 pseudo-rejected samples for training.

**Results.** Tab. 1 reports mean reward-model scores on SD v1.5 across three test sets. TGO achieves strong performance relative to both supervised baselines and prior preference-alignment methods. Additional results are provided in App. C.

**Sensitivity to the percentile threshold.** Because TGO replaces the oracle instance-dependent baseline with an empirical threshold, we study its sensitivity to the percentile choice $p$. Tab. 3 shows that TGO is reasonably stable within

unpaired conversion so that TGO and Diffusion-KTO operate on comparable supervision signals, avoiding confounds from different preprocessing.

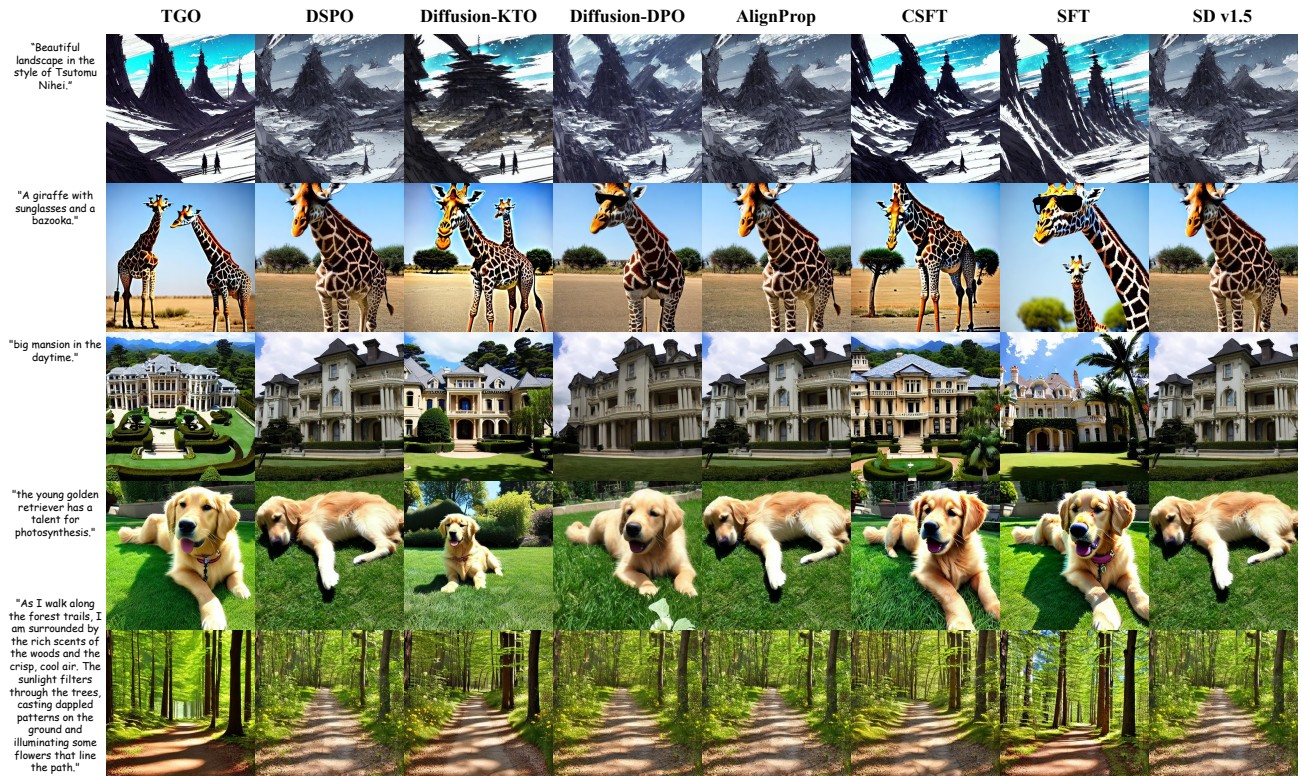

*Figure 2.* Qualitative comparison on Stable Diffusion v1.5. Each column corresponds to one finetuning method (from left to right: TGO (ours), DSPO, Diffusion-KTO, Diffusion-DPO, AlignProp, CSFT, SFT, and the original SD v1.5). Each row shows images generated from the same text prompt (listed on the left), sampled from the HPS v2, Pick-a-Pic, and PartiPrompts test sets.

the practical range $p \in \{0.3, 0.5\}$ across Pick-a-Pic, PartiPrompts, and HPSv2. Very low thresholds include many weak positives, while overly high thresholds produce too few positives and degrade several reward-model scores. We therefore use the median threshold by default, while treating $p$ as a simple validation hyperparameter when a held-out scalar-feedback set is available.

### 4.4. Qualitative Results

Fig. 2 provides side-by-side comparisons on SD v1.5. Across prompts sampled from HPSv2, Pick-a-Pic, and PartiPrompts, TGO better preserves prompt-relevant attributes and reduces obvious artifacts compared with SFT/CSFT and several preference-based baselines. Additional qualitative examples are deferred to App. D.

### 4.5. Text-to-Video Generalization

We further evaluate TGO on text-to-video generation. Starting from VidProM, we construct a 15,218-prompt subset[3]

---

[3]Specifically, we first select prompts with length between 100 and 200 from the original VidProM dataset, and then use Qwen to filter out prompts that are unnatural, incomplete, or grammatically incorrect. This yields 15,218 high-quality prompts.

and split it into training and test sets with an 8:2 ratio. For all experiments, we randomly subsample the training split for computational efficiency.

As the backbone, we adopt Wan 1.3B (Wan et al., 2025) and use VideoReward (Liu et al., 2025b) as the reward model. Tab. 4 reports VideoAlign metrics. Compared with the supervised fine-tuning baseline (SFT-LoRA) and the preference-optimization baseline (KTO-LoRA), TGO-LoRA improves the overall VideoReward score and yields better VQ and MQ while maintaining competitive temporal alignment, suggesting that the threshold-guided scheme naturally extends from images to video generation.

### 4.6. Ablations

We ablate key hyperparameters of the threshold-guided loss, including the diffusion temperature $T$, confidence scaling $c$, preference strength $\beta$. All ablations are conducted on SD v1.5 trained on Pick-a-Pic v2 in App. E.

## 5. Limitations

Our method has several limitations. First, the current formulation uses a single global threshold estimated from empiri-

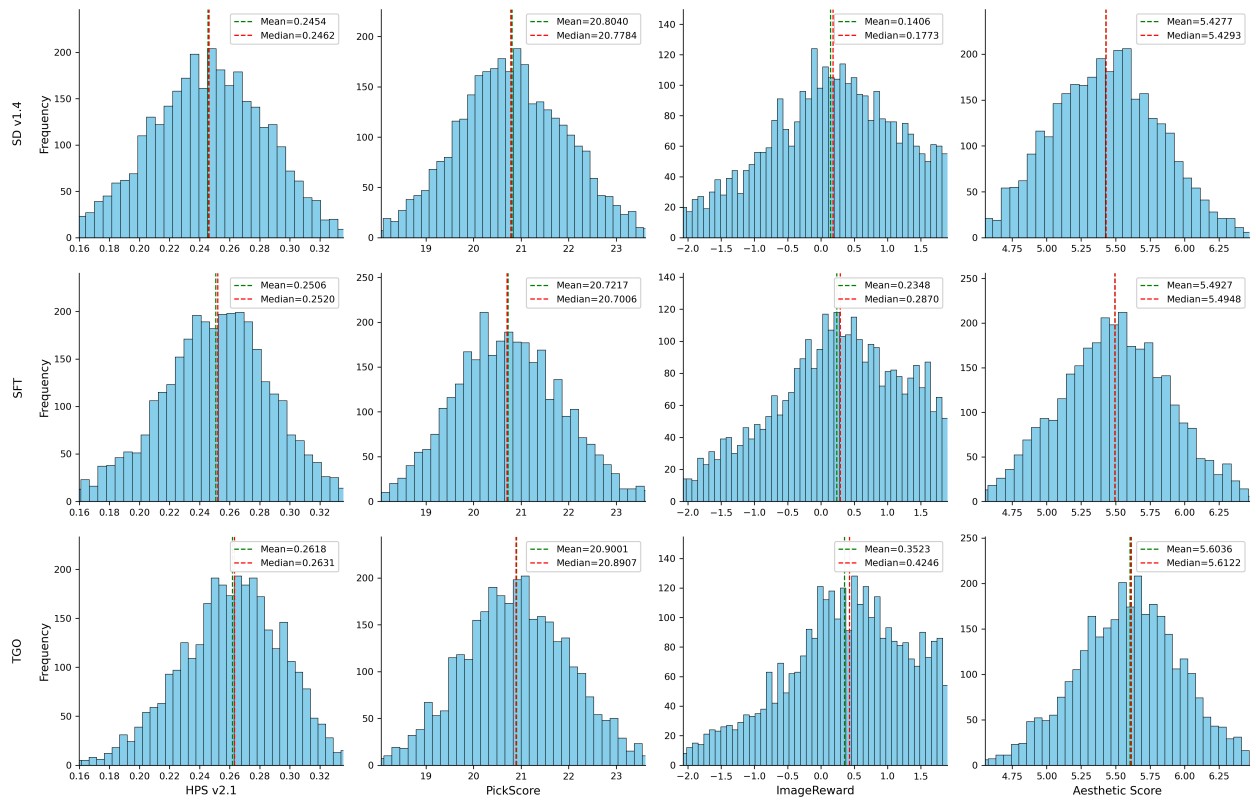

*Figure 3.* Score distributions by model and reward metric for SD v1.4.

*Table 3.* Sensitivity of TGO to the percentile threshold $p$ on Pick-a-Pic v2 training. Higher is better. Best and second-best results are shown in **bold** and underlined.

| $p$ | Pick-a-Pic | | | | | PartiPrompts | | | | | HPSv2 | | | | |
|---|---|---|---|---|---|---|---|---|---|---|---|---|---|---|---|
| | PickScore | HPSv2.1 | CLIPScore | ImageReward | Aesthetic | PickScore | HPSv2.1 | CLIPScore | ImageReward | Aesthetic | PickScore | HPSv2.1 | CLIPScore | ImageReward | Aesthetic |
| 0.1 | 20.93 | 0.2757 | 27.44 | 0.4913 | 5.66 | 21.50 | 0.2751 | 27.14 | 0.5695 | 5.55 | 21.20 | 0.2857 | 29.68 | 0.6720 | 5.59 |
| 0.3 | 21.09 | 0.2841 | 27.62 | **0.6817** | **5.67** | 21.61 | 0.2754 | 27.36 | 0.5859 | **5.59** | 21.37 | **0.2988** | 29.92 | 0.7111 | **5.63** |
| 0.5 | **21.13** | **0.2881** | 27.72 | 0.6728 | 5.63 | **21.62** | **0.2774** | **27.56** | **0.6485** | 5.58 | **21.43** | 0.2935 | 30.17 | **0.7514** | 5.61 |
| 0.7 | 20.97 | 0.2690 | 27.58 | 0.3825 | 5.58 | 21.46 | 0.2616 | 27.33 | 0.4338 | 5.51 | 21.19 | 0.2797 | 30.01 | 0.5171 | 5.59 |
| 0.9 | 20.89 | 0.2660 | **27.82** | 0.3385 | 5.53 | 21.45 | 0.2645 | 27.39 | 0.4643 | 5.43 | 21.26 | 0.2756 | **30.23** | 0.5308 | 5.53 |

*Table 4.* Text-to-video results on the VideoAlign benchmark with Wan 1.3B and VideoReward. TGO-LoRA improves the overall VideoReward score and most component metrics compared with SFT-LoRA and KTO-LoRA.

| Method | VQ Score | MQ Score | TA Score | Overall Score |
|---|---|---|---|---|
| Original | -0.7963 | -0.4316 | -0.8639 | -2.0918 |
| SFT-LoRA | -0.6054 | -0.4159 | **-0.6705** | -1.6918 |
| KTO-LoRA | -0.5832 | -0.1025 | -0.6846 | -1.3703 |
| TGO-LoRA | **-0.5631** | **0.0627** | -0.6753 | **-1.1757** |

cal score statistics. Although this works well in our experiments, it may be suboptimal when score distributions are strongly heterogeneous across prompts or semantic groups, in which case prompt-conditional thresholds could be more appropriate. Second, TGO does not eliminate the dependence on feedback quality: if the scalar scores are noisy, biased, or only imperfectly aligned with human preference,

the induced pseudo-labels can inherit these imperfections.

## 6. Conclusion

In this work, we introduced a threshold-guided optimization framework for aligning visual generative models directly from scalar feedback, without requiring explicitly paired preferences. By revisiting the KL-regularized objective, we interpret alignment as comparing each sample's reward to an intractable instance-specific baseline, and replace this oracle with a global, data-driven threshold combined with a confidence-weighted surrogate loss. Experiments across diffusion and masked generative models, multiple datasets, and diverse reward models show consistent improvements over supervised fine-tuning and recent preference-based baselines. Our work indicates that scalar scores are sufficient to recover much of the benefit of pairwise preference optimization in practical visual generation settings.

## Impact Statement

This paper studies preference optimization for visual generative models using scalar feedback via a threshold-guided objective. The primary intended impact is to reduce the dependence on explicitly paired comparisons, which can be costly to collect and difficult to scale, thereby making alignment pipelines for image/video generation more accessible and efficient.

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

**Main Content.**

## A. Proof of Monotonicity for the Policy Ratio

**Theorem A.1** (Monotonicity of the Policy Ratio). *Let the optimal policy $\pi^*(y|x)$ be defined by the KL-regularized objective:*

$$\pi^*(y|x) = \frac{1}{Z(x)} \pi_{\mathrm{ref}}(y|x) \exp\left(\frac{1}{\beta} r(x,y)\right), \tag{12}$$

*where the partition function $Z(x) = \sum_{y' \in \mathcal{Y}} \pi_{\mathrm{ref}}(y'|x) \exp\left(\frac{1}{\beta} r(x,y')\right)$. Then, for any $y_k \in \mathcal{Y}$ such that $\pi_{\mathrm{ref}}(y_k|x) > 0$, the ratio $\frac{\pi^*(y_k|x)}{\pi_{\mathrm{ref}}(y_k|x)}$ is a strictly increasing function of its reward $r(x,y_k)$, provided there exists at least one alternative response $y' \neq y_k$ with $\pi_{\mathrm{ref}}(y'|x) > 0$.*

*Proof.* Fix $y_k \in \mathcal{Y}$ and define $r := r(x,y_k)$ to simplify notation. We analyze the function:

$$f(r) := \frac{\pi^*(y_k|x)}{\pi_{\mathrm{ref}}(y_k|x)} = \frac{\exp\left(r/\beta\right)}{Z(x)}, \tag{13}$$

where the normalization constant $Z(x)$ depends on $r$, since $r(x,y_k)$ is one of the terms in its summation.

To determine if $f(r)$ is strictly increasing, we compute its derivative with respect to $r$ using the quotient rule. Let $u(r) := \exp(r/\beta)$ and $v(r) := Z(x)$. Then $\frac{df}{dr} = \frac{u'v - uv'}{v^2}$.

The derivatives of $u(r)$ and $v(r)$ are:

$$u'(r) = \tfrac{1}{\beta} \exp(r/\beta),$$

$$v'(r) = \tfrac{d}{dr}\left[\sum_{y' \in \mathcal{Y}} \pi_{\mathrm{ref}}(y'|x) \exp\left(\tfrac{r(x,y')}{\beta}\right)\right] = \pi_{\mathrm{ref}}(y_k|x) \tfrac{1}{\beta} \exp(r/\beta). \tag{14}$$

The derivative $v'(r)$ only contains the term corresponding to $y_k$ because all other rewards $r(x,y')$ for $y' \neq y_k$ are treated as constants with respect to $r$.

Substituting these into the quotient rule expression:

$$\begin{aligned}
\frac{df}{dr} &= \frac{\left(\frac{1}{\beta} e^{r/\beta}\right) Z(x) - e^{r/\beta}\left(\pi_{\mathrm{ref}}(y_k|x) \frac{1}{\beta} e^{r/\beta}\right)}{Z(x)^2} \\
&= \frac{e^{r/\beta}}{\beta Z(x)^2}\left[Z(x) - \pi_{\mathrm{ref}}(y_k|x) e^{r/\beta}\right].
\end{aligned} \tag{15}$$

The term in the brackets simplifies to:

$$\begin{aligned}
Z(x) - \pi_{\mathrm{ref}}(y_k|x) e^{r/\beta} &= \left(\sum_{y' \in \mathcal{Y}} \pi_{\mathrm{ref}}(y'|x) \exp\left(\tfrac{r(x,y')}{\beta}\right)\right) - \pi_{\mathrm{ref}}(y_k|x) \exp\left(\tfrac{r}{\beta}\right) \\
&= \sum_{y' \neq y_k} \pi_{\mathrm{ref}}(y'|x) \exp\left(\tfrac{r(x,y')}{\beta}\right).
\end{aligned} \tag{16}$$

Since $\pi_{\text{ref}}(y'|x) \geq 0$ and $\exp(\cdot) > 0$, each term in this sum is non-negative. By the theorem's condition, there is at least one $y' \neq y_k$ with $\pi_{\text{ref}}(y'|x) > 0$, so this sum is strictly positive.

Therefore, the derivative in Eq. 15 is a product of strictly positive terms:

$$\frac{df}{dr} = \underbrace{\frac{\exp(r/\beta)}{\beta \, Z(x)^2}}_{>0} \cdot \underbrace{\left( \sum_{y' \neq y_k} \pi_{\text{ref}}(y'|x) \, \exp\left(\frac{r(x,y')}{\beta}\right) \right)}_{>0} > 0. \tag{17}$$

Since the derivative is strictly positive, the function $f(r)$ is strictly increasing in $r$. $\qquad \square$

# B. Formal Guarantees for Threshold-Guided Optimization

This appendix collects the formal assumptions, theorems, and proofs for the guarantees of our threshold-guided objective $L_{\text{TG}}$ summarized in Theorem 3.1 in the main text. We view the minimizer of $L_{\text{TG}}$ as the *threshold-guided estimator* (TGO estimator).

## B.1. Assumptions

**Assumption B.1** (Regularity Conditions for Threshold-Guided Optimization). Let $\ell(\theta; z)$ denote the per-sample threshold-guided loss (the contribution of a single sample $z$ to $L_{\text{TG}}$). Assume:

1. (**Identifiability**) The population loss $L(\theta) = \mathbb{E}_z[\ell(\theta; z)]$ has a unique minimizer $\theta^*$ in an open neighborhood $\mathcal{N}$.

2. (**Smoothness**) $\ell(\theta; z)$ is three-times continuously differentiable in $\mathcal{N}$ almost surely. The population derivatives $\nabla^k L(\theta)$ for $k = 1, 2, 3$ exist and are continuous at $\theta^*$.

3. (**Regularity**) The Hessian $H = \nabla^2 L(\theta^*)$ is positive definite. The score $\nabla\ell(\theta^*; z)$ has finite second moments with covariance $S = \text{Cov}(\nabla\ell(\theta^*; z))$. A Central Limit Theorem holds for $\sqrt{n} \, \nabla L_n(\theta^*)$.

## B.2. Consistency of the Threshold-Guided Estimator

**Corollary B.2** (Consistency of Threshold-Guided Optimization). *Under Assumption B.1, the empirical TGO estimator*

$$\hat{\theta}_n = \arg\min_\theta L_n(\theta), \quad L_n(\theta) = \frac{1}{n} \sum_{i=1}^{n} \ell(\theta; z_i)$$

*is consistent, i.e., $\hat{\theta}_n \xrightarrow{p} \theta^*$ as $n \to \infty$.*

*Proof.* By standard M-estimation theory, $L_n(\theta)$ converges uniformly to $L(\theta)$, and $L(\theta)$ has a unique minimizer $\theta^*$ under Assumption B.1. The argmin consistency theorem therefore yields $\hat{\theta}_n \xrightarrow{p} \theta^*$. $\qquad \square$

## B.3. Asymptotic Bias of the Threshold-Guided Estimator

**Theorem B.3** (Asymptotic Bias of Threshold-Guided Optimization). *Under Assumption B.1, the expectation of $\hat{\theta}_n$ admits the expansion*

$$\mathbb{E}[\hat{\theta}_n] - \theta^* = \frac{1}{n} B_1(\theta^*) + o(1/n),$$

*where the a-th coordinate of $B_1(\theta^*)$ is*

$$(B_1(\theta^*))_a = -\frac{1}{2} H_{ab}^{-1} J_{bcd} (H^{-1} S H^{-1})_{cd}, \tag{18}$$

*with $H = \nabla^2 L(\theta^*)$, $S = \text{Cov}(\nabla\ell(\theta^*; z))$, and $J_{bcd} = \mathbb{E}[\partial^3 \ell(\theta^*; z)/\partial\theta_b \partial\theta_c \partial\theta_d]$.*

*Proof.* **Step 1: First-order condition and Taylor expansion.** By optimality, $\nabla L_n(\hat{\theta}_n) = 0$. Expanding around $\theta^*$ gives

$$0 = \nabla L_n(\theta^*) + \nabla^2 L_n(\theta^*)(\hat{\theta}_n - \theta^*)$$
$$+ \tfrac{1}{2}\nabla^3 L_n(\bar{\theta})[\hat{\theta}_n - \theta^*, \hat{\theta}_n - \theta^*] + r_n, \tag{19}$$

where $\bar{\theta}$ lies between $\hat{\theta}_n$ and $\theta^*$, and $r_n = o_p(\|\hat{\theta}_n - \theta^*\|^2) = o_p(n^{-1})$.

**Step 2: Isolate $\Delta = \hat{\theta}_n - \theta^*$.** Rearranging Eq. 19 yields

$$\Delta = -\left[\nabla^2 L_n(\theta^*)\right]^{-1}\nabla L_n(\theta^*)$$
$$- \tfrac{1}{2}\left[\nabla^2 L_n(\theta^*)\right]^{-1}\nabla^3 L_n(\bar{\theta})[\Delta, \Delta] + o_p(n^{-1}). \tag{20}$$

**Step 3: Take expectations.** Since $\mathbb{E}[\nabla L_n(\theta^*)] = 0$ and $\nabla^2 L_n(\theta^*) \xrightarrow{p} H$, we can replace the empirical Hessian and third derivatives by their population counterparts $H$ and $J$ up to $o(n^{-1})$ terms:

$$\mathbb{E}[\Delta] = -\tfrac{1}{2}H^{-1}J\,\mathbb{E}[\Delta \otimes \Delta] + o(n^{-1}). \tag{21}$$

**Step 4: Insert asymptotic covariance.** From standard M-estimator theory,

$$\mathbb{E}[\Delta \otimes \Delta] = \frac{1}{n}H^{-1}SH^{-1} + o(n^{-1}). \tag{22}$$

Substituting Eq. 22 into Eq. 21 gives

$$\mathbb{E}[\Delta] = -\frac{1}{2n}H^{-1}J\left(H^{-1}SH^{-1}\right) + o(n^{-1}),$$

which matches Eq. 18. $\qquad\square$

**Remark.** If $\ell(\theta; z)$ is the negative log-likelihood of a correctly specified model, then $S = H = I(\theta^*)$ (the Fisher information), which further simplifies the bias term.

### B.4. Calibration of Threshold-Guided Pseudo-Labels

**Proposition B.4** (Calibration of Threshold-Guided Pseudo-Labels)**.** *Let $\tau^*(x) = \beta \log Z(x)$ denote the KL-optimal baseline. Suppose scores satisfy $s = g(R(x,y)) + \xi$, where $g$ is strictly increasing and $\xi$ is sub-Gaussian. If $\tau$ is estimated as the empirical p-quantile with error $\varepsilon_\tau = O(1/\sqrt{n})$, then*

$$\Pr\left[l \neq l^*\right] = O(\varepsilon_\tau + \|\xi\|_{\psi_2}),$$

*where $l = \mathbb{1}[s \geq \tau]$ and $l^* = \mathbb{1}[R(x,y) \geq \tau^*(x)]$.*

*Proof.* By Theorem A.1 in Appendix A, the KL-optimal rule is equivalent to thresholding $R(x,y)$ against $\tau^*(x)$. Since $g$ is strictly monotone, comparing $s$ is equivalent to comparing $R$ up to the additive noise $\xi$. The empirical quantile $\tau$ concentrates around the true quantile at rate $O(1/\sqrt{n})$, and label flips occur only when either $\xi$ or $\varepsilon_\tau$ is large enough to cross the decision boundary. This yields the stated bound. $\qquad\square$

## C. Additional Quantitative Results on Text-to-Image Generation

We further conduct a GPT-based evaluation of threshold-guided optimization over Stable Diffusion v1.5. For each prompt in the Pick-a-Pic test set, we prepare one image generated by TGO and one image generated by a baseline model, and then randomly swap their order with 50% probability. We adopt GPT-5 as an automated judge with the following comparative instruction: *"Given the text, which image better matches the description in terms of semantics and visual quality?"* For each baseline, we report the fraction of prompts where GPT prefers TGO, as well as the tie rate, in Figure 4. GPT preferences are consistent with the reward-model–based metrics in Tab. 1 and consistently favor TGO over all baselines.

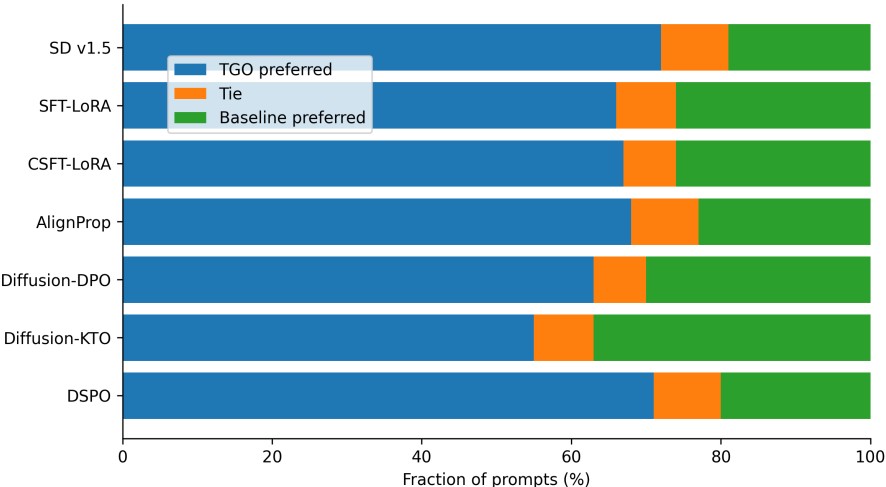

*Figure 4.* GPT-based evaluation of threshold-guided optimization over Stable Diffusion v1.5 on the Pick-a-Pic test set. For each baseline, we show a stacked horizontal bar indicating the fraction of prompts where GPT-5 prefers TGO, judges a tie, or prefers the baseline.

*Table 5.* Ablation of TGO hyperparameters on Stable Diffusion v1.5: win rate (%) of the baseline configuration ($\beta{=}1, c{=}5, T{=}10^{-3}$) against other variants, evaluated with five reward models.

| Fixed Hyperparameters | Ablation Settings | Win rate of baseline vs. ablations | | | | |
|---|---|---|---|---|---|---|
| | | HPS v2 | PickScore | ImageReward | Aesthetic | CLIP |
| $c{=}10, T{=}10^{-3}$ | $\beta{=}0.1$ | 55.48% | 58.73% | 57.17% | 65.64% | 56.60% |
| | $\beta{=}0.5$ | 54.90% | 58.49% | 57.54% | 65.97% | 53.30% |
| | $\beta{=}1$ | 100.00% | 100.00% | 100.00% | 100.00% | 100.00% |
| | $\beta{=}5$ | 60.85% | 60.38% | 53.30% | 54.25% | 50.00% |
| | $\beta{=}10$ | 99.53% | 89.86% | 98.82% | 97.64% | 99.29% |
| $\beta{=}1, T{=}10^{-3}$ | $c{=}1$ | 66.87% | 66.54% | 65.45% | 59.49% | 67.09% |
| | $c{=}2$ | 66.73% | 66.03% | 63.09% | 59.09% | 67.19% |
| | $c{=}5$ | 100.00% | 100.00% | 100.00% | 100.00% | 100.00% |
| | $c{=}10$ | 45.28% | 63.02% | 51.95% | 55.43% | 55.71% |
| | $c{=}20$ | 45.28% | 60.87% | 51.30% | 54.27% | 51.01% |
| $\beta{=}1, c{=}10$ | $T{=}10^{-1}$ | 57.46% | 76.79% | 59.67% | 68.16% | 75.08% |
| | $T{=}10^{-2}$ | 59.48% | 76.32% | 60.07% | 69.36% | 74.13% |
| | $T{=}10^{-3}$ | 100.00% | 100.00% | 100.00% | 100.00% | 100.00% |
| | $T{=}10^{-4}$ | 59.20% | 62.26% | 54.25% | 55.66% | 55.66% |
| | $T{=}10^{-5}$ | 58.25% | 66.27% | 55.19% | 56.13% | 55.42% |

## D. Additional Qualitative Results on Text-to-Image Generation

We also provide additional qualitative examples for text-to-image generation. Figures 5 and 6 show further side-by-side comparisons of images generated by SD v1.5 fine-tuned with different alignment methods.

Figures 7, 8, 9 and 10 show analogous side-by-side comparisons for Meissonic and Flux fine-tuned with different alignment methods.

Across all comparison figures, TGO consistently outperforms both supervised fine-tuning and prior preference-alignment baselines, producing images that better match the prompts in both semantics and visual quality.

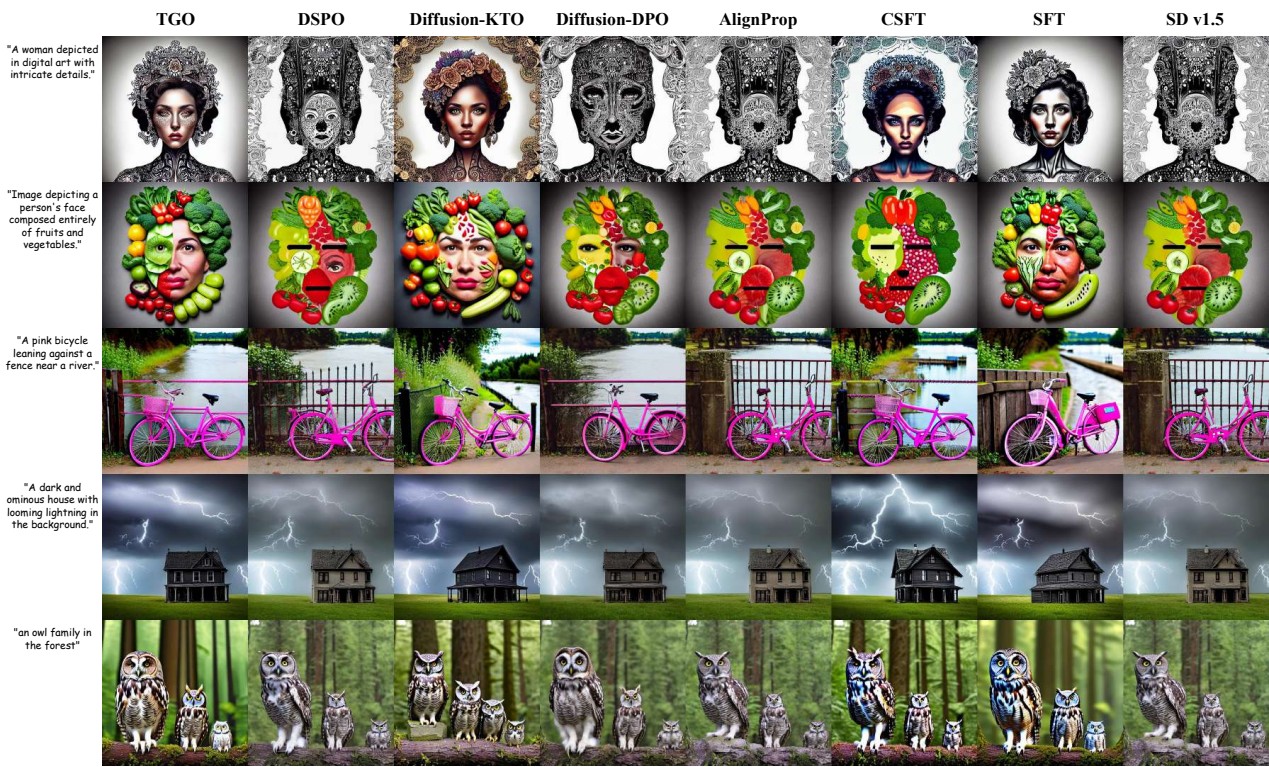

*Figure 5.* More qualitative comparison on Stable Diffusion v1.5. Each column corresponds to one finetuning method (from left to right: TGO (ours), DSPO, Diffusion-KTO, Diffusion-DPO, AlignProp, CSFT, SFT, and the original SD v1.5). Each row shows images generated from the same text prompt (listed on the left), sampled from the HPS v2, Pick-a-Pic, and PartiPrompts test sets.

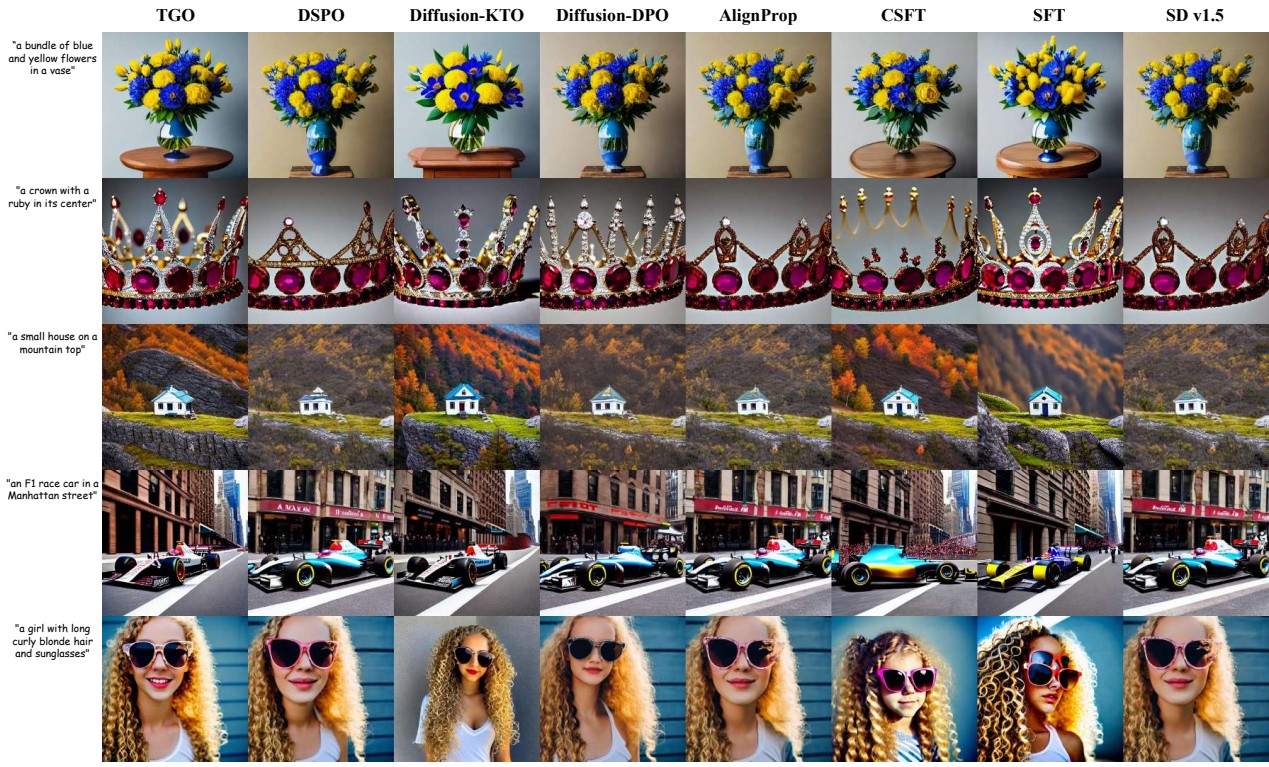

*Figure 6.* More qualitative comparison on Stable Diffusion v1.5. Each column corresponds to one finetuning method (from left to right: TGO (ours), DSPO, Diffusion-KTO, Diffusion-DPO, AlignProp, CSFT, SFT, and the original SD v1.5). Each row shows images generated from the same text prompt (listed on the left), sampled from the HPS v2, Pick-a-Pic, and PartiPrompts test sets.

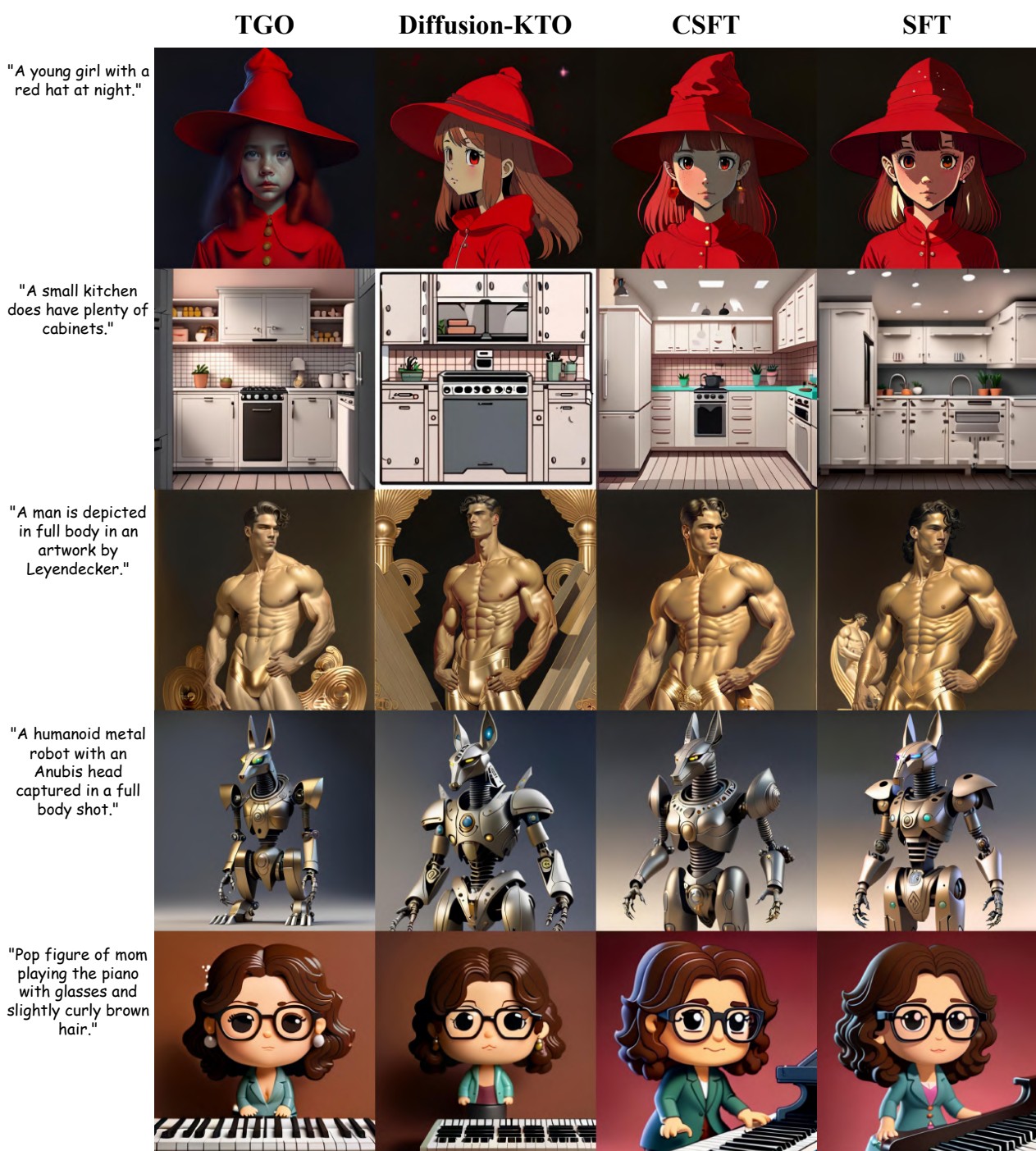

*Figure 7.* More qualitative comparison on Meissonic. Each column corresponds to one finetuning method (from left to right: TGO (ours), Diffusion-KTO, CSFT and SFT). Each row shows images generated from the same text prompt (listed on the left), sampled from the HPS v2, Pick-a-Pic, and PartiPrompts test sets.

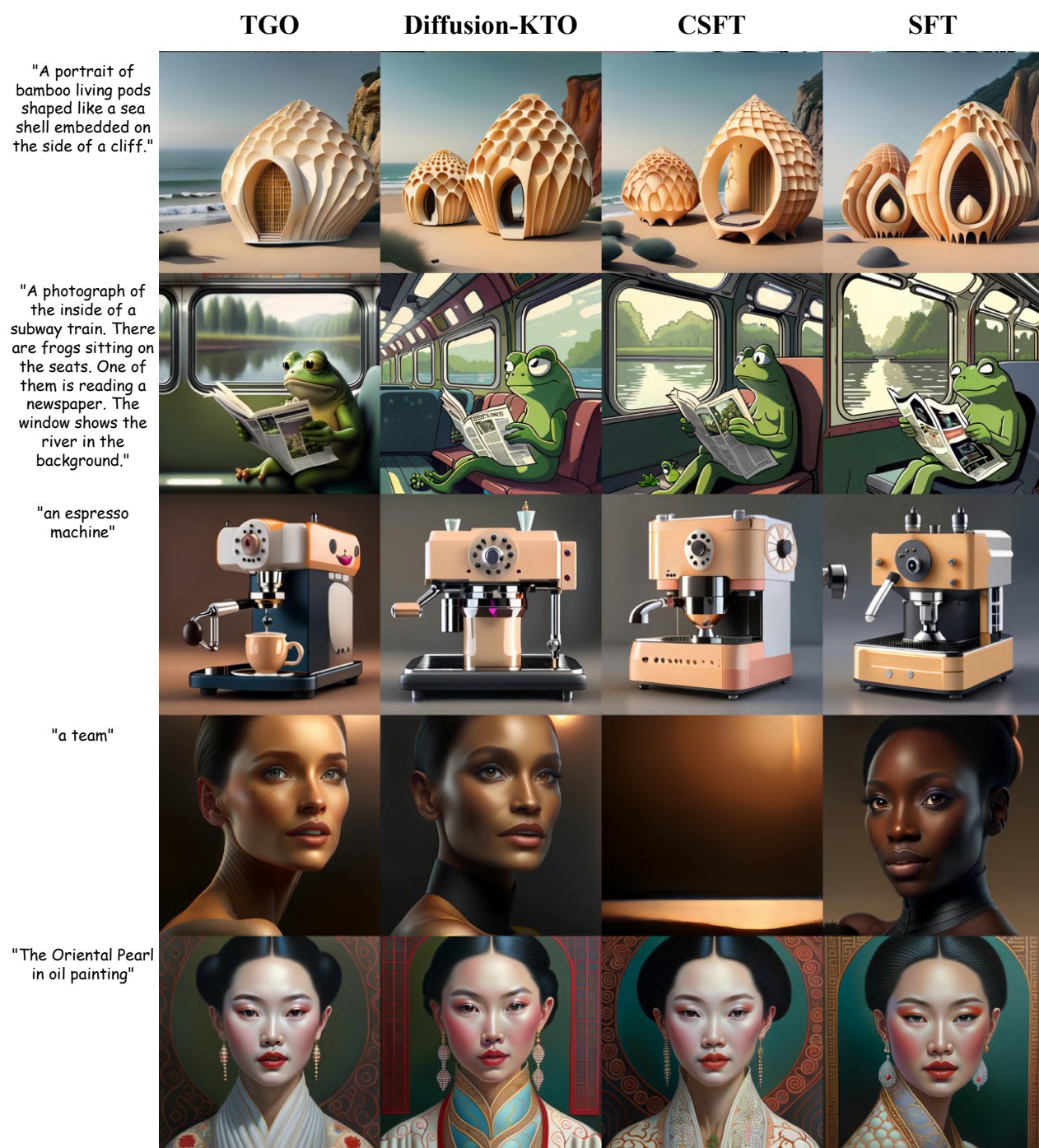

*Figure 8.* More qualitative comparison on Meissonic. Each column corresponds to one finetuning method (from left to right: TGO (ours), Diffusion-KTO, CSFT and SFT). Each row shows images generated from the same text prompt (listed on the left), sampled from the HPS v2, Pick-a-Pic, and PartiPrompts test sets.

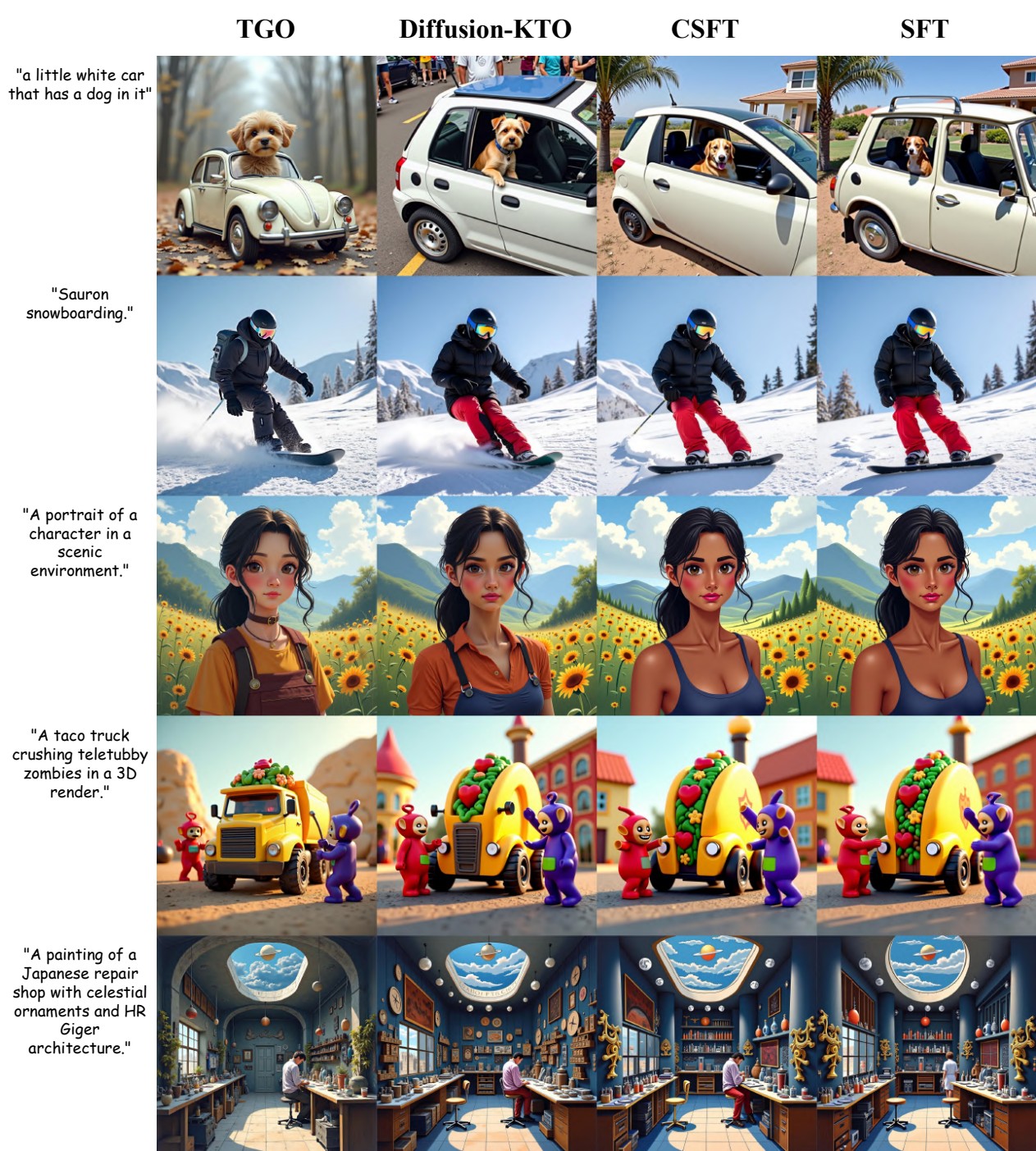

*Figure 9.* More qualitative comparison on Flux. Each column corresponds to one finetuning method (from left to right: TGO (ours), Diffusion-KTO, CSFT and SFT). Each row shows images generated from the same text prompt (listed on the left), sampled from the HPS v2, Pick-a-Pic, and PartiPrompts test sets.

| TGO | Diffusion-KTO | CSFT | SFT |
|-----|---------------|------|-----|

"a book cover"

"hope"

"A tornado made of tigers crashing into a skyscraper. painting in the style of Hokusai."

"a small house on a mountain top"

"a lizard"

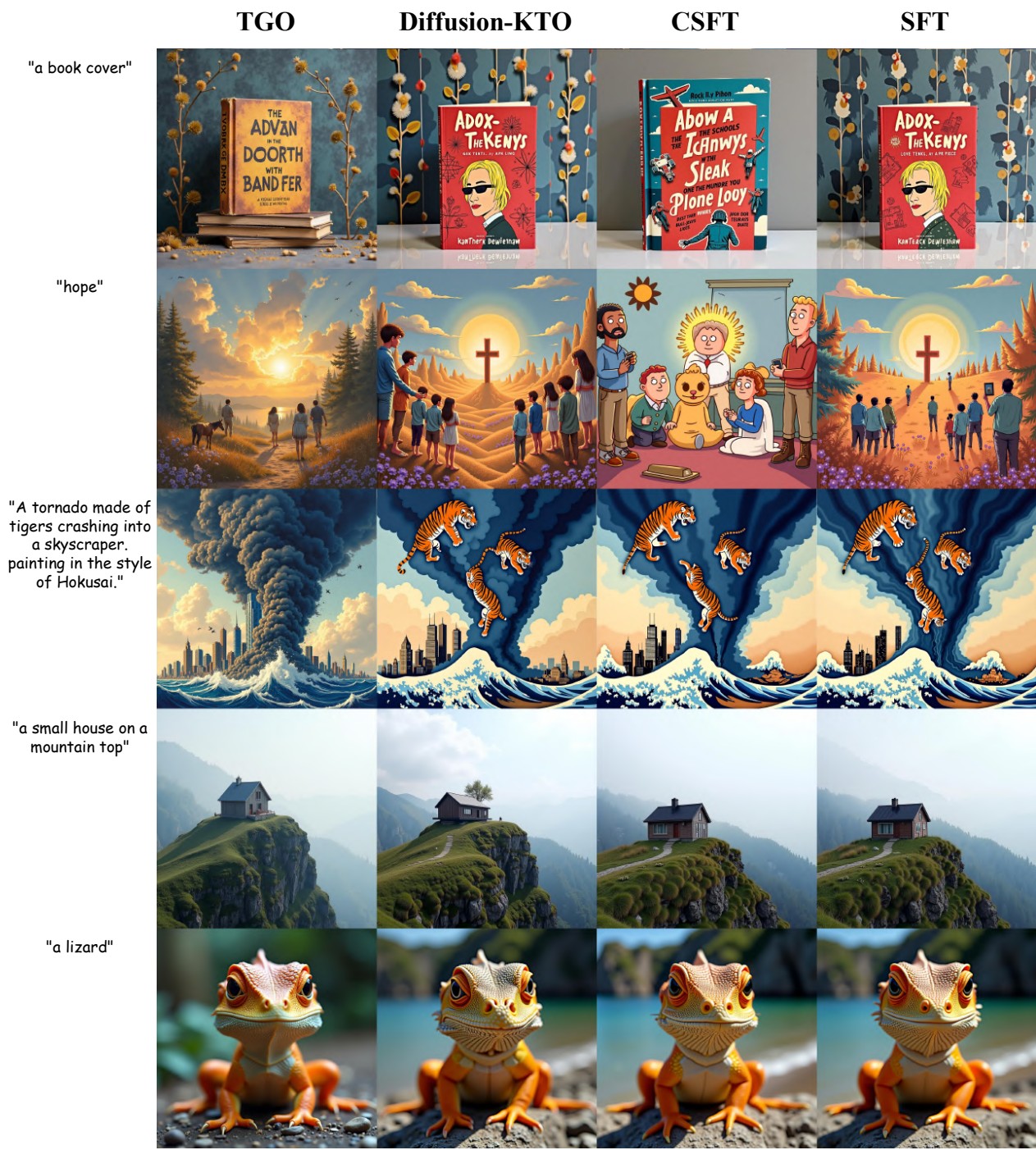

*Figure 10.* More qualitative comparison on Flux. Each column corresponds to one finetuning method (from left to right: TGO (ours), Diffusion-KTO, CSFT and SFT). Each row shows images generated from the same text prompt (listed on the left), sampled from the HPS v2, Pick-a-Pic, and PartiPrompts test sets.

# E. Ablation Study

In this section, we ablate the key hyperparameters of our threshold-guided loss: the temperature $T$ used in the diffusion log-likelihood approximation, the difference scaling factor $c$, and the preference strength $\beta$. All experiments are conducted on Stable Diffusion v1.5 with Pick-a-Pic v2. We present win rates of different combinations of key hyperparameters in Tab. 5. Overall, for SD v1.5 the best TGO hyperparameter setting is $\beta = 1$, $c = 5$, and $T = 0.001$. Other foundation models may favor different settings, but this combination serves as a strong default in practice.

# F. Pseudocode of the Threshold-Guided Loss

We provide PyTorch-style pseudocode for the threshold-guided loss. The implementation closely follows the formulation in Section 3.5, using log-likelihood ratios between the current policy and the reference policy, reweighted by relative scores:

```python
import torch
import torch.nn.functional as F

def compute_tgo_loss(log_probs, ref_log_probs, relative_scores,
                     beta, c):
    """Args:
        log_probs: log pi_theta(y|x), shape (B,)
        ref_log_probs: log pi_ref(y|x), shape (B,)
        relative_scores: r(y) - tau, shape (B,)
        beta: temperature for reward difference
        c: weight scaling for RM difference
    """
    # Implicit reward difference induced by the policy ratio
    reward_diff = beta * (log_probs - ref_log_probs) # (B,)

    # Preference direction and confidence weighting
    signs = torch.sign(relative_scores) # (B,)
    weights = 1 + c * relative_scores.abs() # (B,)

    # Binary logistic loss (use log1p for numerical stability)
    sig = torch.sigmoid(reward_diff)
    pos_loss = -weights * torch.log(sig + 1e-12)
    neg_loss = -weights * torch.log1p(-sig + 1e-12)

    loss = torch.where(signs >= 0, pos_loss, neg_loss)
    return loss.mean()
```

