# OpenReview forum: "Threshold-Guided Optimization for Visual Generative Models"
_ICML.cc/2026/Conference — ICML 2026 regular_

### Official Review · Reviewer_GEY2 · 2026-02-15

**Soundness:** 3
**Presentation:** 3
**Significance:** 3
**Originality:** 3
**Overall Recommendation:** 4
**Confidence:** 4

**Summary:**

This paper proposes the TGO algorithm to optimize policies from a novel perspective on diffusion preference alignment. It introduces \beta * logz(x) as a decision threshold and formulates the alignment problem as a binary classification task, providing a new viewpoint in the modern reinforcement learning era. Experimental results demonstrate the effectiveness of the proposed method.

**Compliance With Llm Reviewing Policy:**

Affirmed.

**Final Justification:**

This paper approaches preference learning from a novel perspective. In addition, the authors responded constructively to the concerns I raised. Therefore, taking into account the paper’s strengths, weaknesses, and the authors’ rebuttal, I would give this paper a weak accept.

**Key Questions For Authors:**

See weakness.

**Limitations:**

See weakness.

**Strengths And Weaknesses:**

**Strengths**
- This paper presents a novel perspective on preference learning by formulating it as a binary classification problem.
- The presentation of this paper is clear and easy to follow.
- The experimental results demonstrate the effectiveness of the proposed method.

**Weakness**
- The argument in the introduction that “in many practical settings, especially for visual generative models, feedback is more naturally collected as unpaired samples with scalar scores” may not be entirely accurate. In many applications, pairwise comparison is considered more stable and has been widely adopted, particularly in areas such as low-level vision and visual quality assessment. In contrast, single-stimulus rating is often less reliable, as discussed in prior studies. For example, when subjects begin rating the first group of images, they lack a clear internal reference, which can lead to high uncertainty and inconsistent scoring. As a result, scalar ratings may not always reflect perceptual differences as robustly as comparative judgments.
- In fig2, the images optimized by TGO appear to lack naturalness and seem over-optimized, which may indicate a form of reward hacking. This phenomenon should be analyzed in greater detail.

Overall, the paper presents a novel idea. I would therefore lean toward a weak accept.

---

> ### Author Rebuttal · Authors · 2026-03-31
>
> We sincerely thank the reviewer for the constructive feedback. We address each point below.
>
> **W1: The argument in the introduction.**
>
> Thanks for your thoughtful comment. We agree that the current Introduction is too strong. Our intent was not to claim that scalar ratings are universally more reliable than pairwise comparisons, or that pairwise feedback is not widely used. In fact, we agree that pairwise judgments are often more stable in perceptual evaluation settings. Our point is narrower: in many practical visual-generation pipelines, feedback is also frequently available in the form of independent scalar signals, such as star ratings, user engagement signals, or scores from existing evaluators, and these signals can be easier to collect or reuse at scale than explicit pairwise annotations. We will revise the Introduction to make this distinction clearer and soften the wording from "more naturally collected" to a more precise statement.
>
> **W2: Concern about over-optimization.**
>
> Thanks for your thoughtful comment. We agree that the qualitative evidence in Fig. 2 is not sufficient by itself to rule out over-optimization or reward hacking, and that some examples may appear less natural depending on the viewer’s preference. We will revise this section to provide a more careful, example-by-example analysis of where TGO improves. At the same time, we would like to clarify that our conclusion is not based on Fig. 2 alone. In the paper, we evaluate across three benchmark sets and multiple reward models specifically to reduce dependence on any single scorer, and we also include a GPT-based analysis in Fig. 5 to examine whether the gains reflect broader improvements rather than reward hacking.
>
> Additionally, to address the concern about reward hacking, we additionally report cross-reward evaluation for models trained with HPS-based feedback and evaluated using four different reward models:
>
> | | HPSv2.1 Mean ↑ | HPSv2.1 Median ↑ | PickScore Mean ↑ | PickScore Median ↑ | ImageReward Mean ↑ | ImageReward Median ↑ | Aesthetic Mean ↑ | Aesthetic Median ↑ |
> |---|---:|---:|---:|---:|---:|---:|---:|---:|
> | SD v1.4 | 0.2454 | 0.2462 | 20.8040 | 20.7784 | 0.1406 | 0.1773 | 5.4277 | 5.4293 |
> | +SFT | 0.2506 | 0.2520 | 20.6996 | 20.6706 | 0.3136 | 0.3741 | 5.4475 | 5.4462 |
> | +TGO | 0.2618 | 0.2631 | 20.9047 | 20.8774 | 0.4233 | 0.4948 | 5.4919 | 5.4916 |
>
> The key observation is that TGO improves consistently across all four evaluators, rather than only on a single metric. Compared with the baseline model and SFT, TGO achieves better scores under all four reward models. Therefore, these cross-reward results substantially alleviate the concern that TGO's gains are due to reward hacking, and instead reflect better overall generation quality.
>
> These results suggest that the improvement are not merely tied to a specific reward signal. In the revision, we will make this point more explicit.

---

> > ### Author Rebuttal · Reviewer_GEY2 · 2026-04-01
> >
> > The author has addressed my concerns, so I will keep my original score.

---

> > > ### Author Response · Authors · 2026-04-07
> > >
> > > We are glad that our rebuttal has adequately addressed the reviewer's concerns. We sincerely thank the reviewer for the constructive engagement throughout the discussion. The discussion and feedback has been very helpful and has greatly strengthened the paper.

---

### Official Review · Reviewer_Dc1G · 2026-02-20

**Soundness:** 3
**Presentation:** 2
**Significance:** 3
**Originality:** 2
**Overall Recommendation:** 4
**Confidence:** 3

**Summary:**

The paper introduces Threshold-Guided Optimization (TGO), an alignment framework for visual generative models that utilizes unpaired scalar feedback. The authors propose a data-driven global threshold to convert scalar scores into binary "pseudo-preferences," bypassing the need for explicit paired data required by methods like DPO. They provide a theoretical grounding by relating their thresholding mechanism to an approximation of the KL-optimal decision rule. Experiments are conducted across diffusion and masked generative models, including extensions to text-to-video generation.

**Compliance With Llm Reviewing Policy:**

Affirmed.

**Final Justification:**

I suggest accept if the rest of the reviewers feel strongly about it. The rebuttal answered my questions but I remain a bit hesitant about the amount of novelty for ICML standards, even after the clarification from the authors.

**Key Questions For Authors:**

Please see weakness section

**Limitations:**

yes

**Strengths And Weaknesses:**

**Strengths**:

- Mathematically Sound: The paper provides a principled derivation of the TGO framework as a tractable approximation to the KL-optimal decision rule. The theoretical analysis includes proofs of consistency and calibration of the pseudo-labels in the appendix.

- Evaluation on Video: It is nice that the authors demonstrate the versatility of the method by applying it to text-to-video generation.

- Broad Experimental Scope: The method is tested across multiple foundation models (Stable Diffusion, Meissonic, FLUX) and five different reward models, suggesting good generalizability.

**Weaknesses / Areas for Improvement**:

- Conceptual Simplicity & robustness: While the method is effective, the core idea, i.e. using a median/percentile threshold to create binary labels, appears quite simple. The authors should further justify whether this simplicity is a feature (e.g., for stability) or if it limits the model's ability to capture nuanced preferences compared to continuous reward modeling. The reliance on a global data-driven threshold (e.g., p=50) raises concerns about reliability. It is unclear how the method performs when the reward distribution is highly skewed or has high variability. If the dataset consists of mostly "very good" or "very bad" samples, a fixed percentile might lead to noisy pseudo-labels. I am not convinced about this approach in the "real world" and with noisier/skewed data. Have the authors tested the sensitivity of the model to different threshold percentiles (p)? How does the performance change if the data is intentionally skewed?

- More threshold concerns: The threshold is estimated from "empirical score statistics," using the median. It is unclear if this threshold is updated dynamically as the model improves during training. If the threshold is static, the model may quickly reach a point where most of its new generations are "pseudo-positive," leading to a plateuing effect or gradient vanishing

- Organization and Clarity:

There are instances where equations are mentioned before they are defined. For example, the KL-regularized alignment objective (Eq. 1) is referenced in the Introduction but only explicitly introduced in Section 3.1. This disrupts the flow of the paper and should be checked.

Several paragraphs suffer from bad formatting where the last line of a paragraph is pushed to the next page, please revise this.

Figure 3, which visualizes score distributions, is too small in the current layout, making the axes nearly impossible to read.

- Statistical Significance: The quantitative results in the tables lack standard deviations. Without these metrics, it is difficult to determine if the reported gains (e.g., 21.45 vs 21.24) are statistically significant across different seeds.

- Qualitative Analysis: The qualitative results presented are not immediately or obviously superior to the baselines. The authors need to provide a more detailed analysis or human evaluation to justify why the TGO-generated images are "better" beyond just higher RM scores, which are known to be susceptible to reward hacking.

- Applicability to Text: Could this framework be applied to LLMs? While the paper focuses on visual models, the underlying KL-objective is universal. Please explain if or how TGO could handle text-based scalar feedback (e.g., 1-5 star ratings for dialogue).

- Implicit vs. Explicit Reward Modeling: The paper claims TGO is an alternative to explicit reward modeling. However, the experiments use five different pre-trained reward models (HPS, PickScore, etc.) to provide the scalar feedback. This makes TGO essentially a way to use a reward model without the "RL" part, but it doesn't solve the fundamental dependency on a high-quality external reward signal. Can the authors comment on this?

---

> ### Author Rebuttal · Authors · 2026-03-31
>
> We sincerely thank the reviewer for the constructive feedback. We address each point below.
>
> **W1: Sensitivity to threshold.**
>
> We provide an ablation over different percentile choices on Pick-a-Pic v2 training set:
>
> #### Mean Scores on Pick-a-Pic v2.
>
> | |HPSv2.1|PickScore|ImageReward|Aesthetic|CLIPScore|
> |---|---:|---:|---:|---:|---:|
> |p=0.1|0.2757|20.93|0.4913|5.66|27.44|
> |p=0.3|0.2841|21.09|0.6817|5.67|27.62|
> |p=0.5|0.2881|21.13|0.6728|5.63|27.72|
> |p=0.7|0.2690|20.97|0.3825|5.58|27.58|
> |p=0.9|0.2660|20.89|0.3385|5.53|27.82|
>
>
> #### Mean Scores on PartiPrompts.
>
> | |HPSv2.1|PickScore|ImageReward|Aesthetic|CLIPScore|
> |---|---:|---:|---:|---:|---:|
> |p=0.1|0.2751|21.50|0.5695|5.55|27.14|
> |p=0.3|0.2754|21.61|0.5859|5.59|27.36|
> |p=0.5|0.2774|21.62|0.6485|5.58|27.56|
> |p=0.7|0.2616|21.46|0.4338|5.51|27.33|
> |p=0.9|0.2645|21.45|0.4643|5.43|27.39|
>
> #### Mean Scores on HPSv2.
>
> | |HPSv2.1|PickScore|ImageReward|Aesthetic|CLIPScore|
> |---|---:|---:|---:|---:|---:|
> |p=0.1|0.2857|21.20|0.6720|5.59|29.68|
> |p=0.3|0.2988|21.37|0.7111|5.63|29.92|
> |p=0.5|0.2935|21.43|0.7514|5.61|30.17|
> |p=0.7|0.2797|21.19|0.5171|5.59|30.01|
> |p=0.9|0.2756|21.26|0.5308|5.53|30.23|
>
> These results suggest that TGO is reasonably robust within a practical operating range (p=0.3–0.5), while overly high thresholds lead to degraded performance on several reward models.
>
> **W2: Is the threshold updated dynamically?**
>
> In our current experiments, the threshold is fixed during training and is not updated dynamically. We use an offline training setup: the training samples are first collected and scored, and the threshold is then estimated. An online training setup can require threshold updates as the policy distribution shifts. Empirically, we did not observe evidence of training collapse or gradient vanishing under offline training setup. If under a fixed threshold, a large fraction of samples eventually become pseudo-positive, this would indicate that the model has largely saturated relative to the current offline dataset and feedback distribution, rather than a failure mode of the objective itself. In that case, further improvement would likely require refreshed or higher-quality data, rather than simply continuing training on the same fixed dataset. We will add a short discussion to clarify that our current method is primarily designed for offline scalar-feedback alignment, while dynamic threshold updates in online or iterative data-collection settings are an interesting direction for future work.
>
> **W3: Organization and Clarity.**
>
> In the revision, we will revise the Intro to avoid referring to Eq. (1) before it is formally introduced. We will also carefully adjust the formatting throughout the paper to improve readability and enlarge Fig. 3, increase the font size of the axes and labels to make the figure easier to read.
>
> **W4: Statistical Significance.**
>
> In the paper, all methods were evaluated under the same seed to ensure a comparable setup. We additionally report repeated-run statistics with multiple seeds:
>
> | |PickScore|HPSv2.1|CLIPScore|ImageReward|Aesthetic|
> |---|---:|---:|---:|---:|---:|
> |Pick-a-Pic|21.11±0.08|0.2868±0.0012|27.73±0.11|0.6774±0.0140|5.60±0.05|
> |PartiPrompts|21.53±0.04|0.2794±0.0010|27.15±0.19|0.6568±0.0071|5.56±0.05|
> |HPSv2|21.59±0.09|0.2932±0.0012|30.32±0.25|0.7675±0.0109|5.55±0.06|
>
> These results show that TGO exhibits relatively small variance across runs.
>
> **W5: Qualitative Analysis.**
>
> We agree that some of the current qualitative examples are visually close. As we have already included additional comparison examples in App. Figs. 6-9, in the revised version, we will remove less informative examples where the differences are too small, and provide a more detailed analysis of the remaining comparisons, explicitly highlighting where TGO improves over others.
>
> We would also like to clarify that the current manuscript already includes an additional GPT-based evaluation in Fig. 5 as complementary evidence beyond reward-model scores.
>
> **W6: Applicability to Text.**
>
> Yes. The high-level idea would remain similar: estimating a threshold from the scalar score distribution (e.g., 2.9 of 1-5 star ratings), deriving pseudo-positive/pseudo-negative labels relative to that threshold, and optimizing a log-ratio objective with respect to a reference policy.
>
> **W7: Implicit vs. Explicit Reward Modeling.**
>
> Our claim is not that TGO removes dependence on feedback quality altogether; instead, it removes the need for explicit paired preference data. In our setting, TGO operates directly on scalar feedback, which may come from human annotators or from an existing scorer, and converts it into threshold-guided supervision without constructing explicit preference pairs. This is already emphasized in the abstract: "While such approaches are conceptually simple, they fundamentally rely on annotated pairs, limiting scalability in settings where feedback is collected as independent scalar ratings".

---

> > ### Author Rebuttal · Reviewer_Dc1G · 2026-04-01
> >
> > The rebuttal addressed several of my main concerns, especially regarding threshold sensitivity, the offline fixed-threshold setting, and the additional multi-seed evidence. These additions make the empirical case significantly stronger and resolve a number of the issues I had with the original submission.
> >
> > That said, I still remain somewhat hesitant about the overall strength of the contribution relative to the ICML bar. While the results are positive and the method appears practically useful, the core algorithmic idea still feels somewhat limited in novelty and partly heuristic in its final form. In particular, replacing the intractable baseline with a global threshold is simple and effective, but I am not fully convinced that this constitutes a sufficiently strong conceptual advance for a clear acceptance.

---

> > > ### Author Response · Authors · 2026-04-07
> > >
> > > We sincerely thank the reviewer for the thoughtful follow-up. We are glad that the additional analyses helped address several of the original concerns, and we really appreciate the reviewer’s willingness to raise the score after the discussion. The reviewer’s constructive engagement has been very helpful in strengthening the paper.
> > >
> > > We also understand the reviewer’s remaining hesitation regarding novelty. That said, we would respectfully clarify that the contribution is not simply the introduction of a global threshold as a heuristic. The key point is that, under the KL-regularized objective, the ideal policy update for scalar feedback depends on an instance-dependent oracle baseline, which is generally intractable. Our method provides a tractable surrogate for this decision rule in the practically important setting where supervision is available as unpaired scalar scores rather than paired preferences.
> > >
> > > From this perspective, the novelty lies in the reformulation more than in the surface form of the final rule. TGO turns scalar-feedback alignment into a directly optimizable objective without requiring explicit pair construction, and the confidence-weighted term further preserves information from score magnitude rather than reducing supervision to a purely binary signal. We therefore view the method as a simple but meaningful conceptual and practical step for visual generative model alignment under scalar feedback.
> > >
> > > In the revision, we will further clarify this positioning so that the paper does not read as claiming novelty from complexity, but rather from identifying the right optimization paradigm for an underexplored yet practically relevant supervision setting.

---

### Official Review · Reviewer_pP1X · 2026-03-13

**Soundness:** 2
**Presentation:** 3
**Significance:** 3
**Originality:** 3
**Overall Recommendation:** 4
**Confidence:** 4

**Summary:**

This paper focuses on a key limitation of existing DPO-style alignment methods for visual generative models: they rely heavily on pairwise preference data. In real-world image and video generation settings, however, it is often more natural to obtain independent scalar scores for each sample, such as user ratings or outputs from a reward model. To address this, the authors propose Threshold-Guided Optimization (TGO), a method that aligns models using only scalar feedback, without requiring pairwise data. The core idea is to compare each score against a global threshold estimated from the data, and then divide samples into pseudo-positive and pseudo-negative groups. Based on this, training is formulated as a classification problem.

In the experiments, the authors evaluate TGO on several generative models, including Stable Diffusion v1.5, Meissonic, FLUX, and Wan 1.3B, using a range of evaluators such as PickScore, HPSv2.1, ImageReward, CLIPScore, and Aesthetic Score. They report that TGO generally achieves better alignment performance than both conventional supervised fine-tuning and recent preference-based baselines. Furthermore, in text-to-video generation experiments based on Wan 1.3B, the overall VideoReward score also improved, suggesting that this approach can be extended beyond images to video generation as well.

**Compliance With Llm Reviewing Policy:**

Affirmed.

**Final Justification:**

The paper addresses a practically important problem in aligning visual generative models using scalar feedback rather than pairwise preference data, which is a realistic and meaningful setting. I found the core idea of threshold-guided optimization simple, well motivated, and empirically effective.

My main concerns in the original review were about the use of a single global threshold and the lack of comparison to other non-paired or rank-based preference optimization methods. The rebuttal addressed a substantial portion of these concerns.

That said, I still find the theoretical side somewhat less satisfying than the empirical side. The current analysis supports the surrogate objective itself, but the connection to the original alignment objective remains somewhat indirect. For these reasons, I do not view the paper as a completely clear accept.

**Key Questions For Authors:**

- Is it common for the base FLUX model in Table 5 seems to outperform all fine-tuned models? If so, what is the reason for this?
- How sensitive is TGO to the percentile choice \( p \)? Does the method remain stable under different threshold settings?
- Have the authors considered prompt-conditional or cluster-wise thresholds, and if so, how do they compare with the proposed global threshold?
- It would also be helpful to compare TGO against other rank-based preference optimization methods, such as RankDPO, to better demonstrate how effective the proposed objective is in the broader landscape of ranked-feedback alignment.

**Limitations:**

yes

**Strengths And Weaknesses:**

## Strengths

### Soundness
- The paper studies a practically meaningful preference optimization setting based on scalar human feedback rather than paired preference annotations. This is a realistic scenario for modern generative model alignment, where scalar ratings are often easier to collect than pairwise comparisons.
- The proposed surrogate objective is well motivated from the KL-regularized formulation. The derivation is clear, and the paper provides a reasonably structured theoretical analysis of the surrogate estimator, including consistency, asymptotic bias, and a calibration-style connection to the KL-optimal decision rule.

### Significance
- Empirically, the method shows strong performance across multiple models and benchmarks, often outperforming prior DPO-style baselines by a meaningful margin.
- The results suggest that the proposed objective is an effective and scalable alternative for alignment in settings where only scalar feedback is available.

## Weaknesses

### Soundness
- A key concern is the use of a single global data-driven threshold. In practice, the distribution of human scores may vary substantially across prompts, and the alignment between observed scores and latent reward may also be noisier for some prompts than others. Under such heterogeneity, a global threshold could introduce systematic bias, over-emphasize certain prompt types, or lead to uneven alignment quality across prompts.
- The theoretical guarantees mainly concern the surrogate objective rather than the original KL-regularized alignment objective. While the analysis is reasonable for the surrogate estimator itself, the gap between optimizing the surrogate and solving the original KL objective is not rigorously characterized.
- The paper would benefit from additional ablations on the threshold design, such as different percentile choices or prompt-/group-conditional thresholds, since this approximation is central to the method.

### Significance
- A minor empirical concern is that the gains on FLUX appear more marginal than in some other settings (Table 5). In particular, the improvements over strong baselines do not seem uniformly decisive across all alignment metrics.
- Since standard DPO baselines were originally designed for paired preference data, the current comparisons may not fully reflect the strongest alternatives for ranked or scalar-feedback supervision. Including methods such as RankDPO [1] would help position the proposed method more convincingly in the broader landscape of non-paired preference optimization.


[1] Scalable Ranked Preference Optimization for Text-to-Image Generation

---

> ### Author Rebuttal · Authors · 2026-03-31
>
> We sincerely thank the reviewer for the constructive feedback. We address each point below.
>
> **W5 & Q1: Compared to FLUX.**
>
> Thanks for your careful comments. We explained this point in lines 876-879. This is mainly due to the quality of training data. "The Pick-a-Pic training data is of lower quality than the original Meissonic and FLUX outputs. As a result, the win rates of all fine-tuned models against the original checkpoints are below 50%, so we only compare alignment methods against one another here, rather than against the original models." This phenomenon is not specific to TGO: all fine-tuned variants under the Pick-a-Pic-based setting underperform the original FLUX checkpoint, likely due to the training-data quality mismatch.
>
> **W3 & Q2: Sensitivity to threshold.**
>
> We provide an ablation over different percentile choices on Pick-a-Pic v2 training set:
>
> #### Mean Scores on Pick-a-Pic v2.
>
> | |HPSv2.1|PickScore|ImageReward|Aesthetic|CLIPScore|
> |---|---:|---:|---:|---:|---:|
> |p=0.1|0.2757|20.93|0.4913|5.66|27.44|
> |p=0.3|0.2841|21.09|0.6817|5.67|27.62|
> |p=0.5|0.2881|21.13|0.6728|5.63|27.72|
> |p=0.7|0.2690|20.97|0.3825|5.58|27.58|
> |p=0.9|0.2660|20.89|0.3385|5.53|27.82|
>
> Mean scores on PartiPrompts and HPSv2 are provided in our response to Reviewer UBqU Q1. Due to the character limit, we include only the Pick-a-Pic v2 mean scores here.
>
> These results suggest that TGO is reasonably robust within a practical operating range (p=0.3–0.5), while overly high thresholds lead to degraded performance on several reward models.
>
> **W1 & Q3: Prompt-conditional or cluster-wise thresholds.**
>
> We conducted two experiments beyond the proposed global threshold:
>
> 1. Length-bin thresholds. We partitioned prompts into several bins according to prompt length and estimated a separate percentile threshold for each bin.
> 2. Cluster-wise thresholds in semantic space. We encoded prompts using the CLIP text encoder, clustered them with K-means, and estimated one threshold per cluster. We tested multiple granularities, including K = 4, 8, 16.
>
> The results are summarized below.
>
> **Pick-a-Pic results.**
>
> |Method|HPSv2.1↑|PickScore↑|ImageReward↑|Aesthetic↑|CLIPScore↑|
> |---|---:|---:|---:|---:|---:|
> |Global(ours)|0.2868|21.11|**0.6774**|5.60|27.73|
> |Length-bin|0.2867|21.26|0.6639|5.69|28.02|
> |Cluster-wise(K=4)|**0.2869**|21.26|0.6596|**5.70**|28.04|
> |Cluster-wise(K=8)|**0.2869**|**21.27**|0.6535|**5.70**|**28.05**|
> |Cluster-wise(K=16)|0.2868|21.26|0.6607|5.69|28.01|
>
> **PartiPrompts results.**
>
> |Method|HPSv2.1↑|PickScore↑|ImageReward↑|Aesthetic↑|CLIPScore↑|
> |---|---:|---:|---:|---:|---:|
> |Global(ours)|0.2794|21.53|0.6568|5.56|27.15|
> |Length-bin|0.2802|21.59|0.6858|5.64|27.43|
> |Cluster-wise(K=4)|0.2799|21.59|0.6860|5.64|**27.45**|
> |Cluster-wise(K=8)|**0.2801**|**21.60**|**0.6892**|**5.65**|**27.45**|
> |Cluster-wise(K=16)|0.2800|21.59|0.6763|**5.65**|27.43|
>
> **HPSv2 results.**
>
> |Method|HPSv2.1↑|PickScore↑|ImageReward↑|Aesthetic↑|CLIPScore↑|
> |---|---:|---:|---:|---:|---:|
> |Global(ours)|0.2932|21.59|0.7675|5.55|30.32|
> |Length-bin|0.2942|21.79|0.7895|**5.66**|30.79|
> |Cluster-wise(K=4)|0.2942|21.79|**0.7927**|**5.66**|30.78|
> |Cluster-wise(K=8)|**0.2944**|**21.80**|0.7878|5.65|**30.80**|
> |Cluster-wise(K=16)|0.2943|21.79|0.7908|5.65|**30.80**|
>
> From these results, we draw three conclusions. First, length-bin thresholds do not always outperform the global threshold. Second, cluster-wise thresholds are more promising. Third, increasing the clustering granularity does not lead to monotonic improvement.
>
> **W5 & Q4: Compare TGO against RankDPO.**
>
> Following the Syn-Pic protocol, we construct a synthetic preference dataset from 58k prompts in Pick-a-Pic v2 using four base models, yielding 232k image-text pairs. Each pair is scored by five reward models. We normalize the scores from each reward model and compute an weighted average across the five scorers to obtain a final scalar score for each pair. Based on these scores, we apply TGO to estimate a threshold and split the data into pseudo-positive and pseudo-negative samples. Following the RankDPO training setup, we train with a batch size of 1024 for 400 steps. The resulting performance on GenEval is shown below.
>
> |Model|Mean↑|Single↑|Two↑|Counting↑|Colors↑|Position↑|Color Attribution↑|
> |---|---:|---:|---:|---:|---:|---:|---:|
> |SDXL|0.55|0.98|0.74|0.39|0.85|0.15|0.23|
> |SDXL(RankDPO)|0.61|1.00|0.86|0.46|0.90|0.14|0.29|
> |SDXL(Ours)|0.65|0.99|0.85|0.63|0.87|0.25|0.31|
>
>
> **W2: Theoretical guarantees.**
>
> Our current theoretical guarantees are established for the tractable surrogate objective, rather than for the original KL-regularized alignment objective. Our intent is not to claim that optimizing TGO exactly solves the original KL objective. Instead, the theory is meant to show that, under scalar-feedback supervision where the oracle baseline is intractable, TGO provides a statistically well-behaved surrogate with a calibrated connection to the KL-optimal decision rule.

---

> > ### Author Rebuttal · Reviewer_pP1X · 2026-04-04
> >
> > Thank you for the thorough rebuttal and for providing a diverse set of additional experiments and ablation studies. I believe these new results address a substantial portion of the weaknesses that were previously raised, and they make the paper significantly stronger. I still find some aspects of the theory somewhat unsatisfactory or heuristic, so I am still unsure whether this paper should be considered a clear acceptance. Nevertheless, I would appreciate it if the AC could take into account that my overall opinion on this paper is now considerably more positive.
> >
> > PS. I found it quite interesting that in W3&Q2, the results at p=0.7-0.9 appear worse than those at p=0.1. If the authors have any intuition or explanation for this behavior, I would be very interested to hear it.

---

> > > ### Author Response · Authors · 2026-04-07
> > >
> > > We sincerely thank the reviewer for the thoughtful follow-up. We are very glad that the additional experiments and ablations helped address a substantial portion of the original concerns, and we greatly appreciate the reviewer’s note that the overall assessment is now considerably more positive. The reviewer’s constructive feedback has been very helpful in strengthening the paper.
> > >
> > > We also understand the reviewer’s remaining reservation regarding the theory. We agree that the current theoretical results are best viewed as support for the tractable surrogate objective, rather than a complete characterization of the gap to the original KL-regularized objective. More specifically, Appendix A justifies the monotonic decision-rule perspective underlying the threshold formulation, while Appendix B establishes consistency, asymptotic bias, and a calibration-style connection for the surrogate estimator itself. In the revision, we will make this positioning more explicit. Our intent is to provide a tractable surrogate that is theoretically motivated and statistically well-behaved, while being directly usable in the scalar-feedback setting.
> > >
> > > Regarding the reviewer’s question about why p=0.7–0.9 can perform worse than p=0.1, our current intuition is the following. When the percentile is set too high, the pseudo-positive set becomes very sparse, and many moderately good samples are instead treated as pseudo-negative. Since TGO trains on the sign induced by the threshold, this means the model may start suppressing a large number of samples that are still reasonably good, rather than mainly distinguishing clearly good from clearly bad ones. This effect can be amplified by the confidence weighting term, because once the threshold is pushed too far into the upper tail, the optimization becomes increasingly imbalanced toward “pushing down” most of the dataset. By contrast, p=0.1 is also suboptimal, but it retains denser positive supervision and does not misclassify as many moderately good samples as negatives. This is why, empirically, overly high thresholds appear more harmful than moderately low ones.
> > >
> > > Overall, we are sincerely encouraged that the reviewer now views the paper substantially more positively. If the reviewer feels that the discussion has resolved a sufficient portion of the original concerns, we would be grateful if the score could be updated to reflect this more positive assessment.

---

### Official Review · Reviewer_qAQF · 2026-03-13

**Soundness:** 4
**Presentation:** 3
**Significance:** 4
**Originality:** 4
**Overall Recommendation:** 4
**Confidence:** 4

**Summary:**

This paper proposes a method which can be used with unpaired scored data for alignment while regular alignment works still requires pair wise preference data. This work shows that the KL based alignment problem can be reinterpreted and formulated to work with a decision rule that decides whether the model policy should to increase or  decrease the probability of  a particular sample by comparing with a  score threshold calculated over whole dataset.

**Compliance With Llm Reviewing Policy:**

Affirmed.

**Final Justification:**

The authors' response resolved most of my concerns, so I maintain my original positive score.

**Key Questions For Authors:**

1. In table 1, consistently diffusion-KTO is the 2nd winner and TGO winning  over it only marginally. Then is this result depend upon how the pref-to-scalar conversion is done? Why is Diffusion-KTO missing from the related works? It would be good to do a thorough discussion on how this two approaches differ and perform a  sensitivity analysis on different types of conversion schemes, or over the percentile threshold score.

**Limitations:**

Please include a "Limitations" section for the proposed work.

**Strengths And Weaknesses:**

### Strengths

* The paper is well easy to follow and well written.
* The proposed method is novel and theoretically grounded.
* It's nice to see an intuitive utilization of the partition function; which otherwise considered problematic and mostly avoided.

### Weaknesses

* My major concern is related to the evaluation section. TGO performs worse compared to flux (Table 5) consistently across all prompt test sets. Why? Why other baselines perfomed worse when implemented on flux? Does this mean TGO's performance gain is limited and visible when applied with a weaker model like sd1.5?
*  In table 2, are models trained and tested on the same reward model? If yes, in that case, how do we know TGO's gains are not coming from reward hacking? Usually cross reward results should also be shown to eliminate that possibility.

---

> ### Author Rebuttal · Authors · 2026-03-31
>
> We sincerely thank the reviewer for the constructive feedback. We address each point below.
>
> **W1: Compared to FLUX.**
>
> Thanks for your careful comments. We explained this point in lines 876-879. This is mainly due to the quality of training data. "The Pick-a-Pic training data is of lower quality than the original Meissonic and FLUX outputs. As a result, the win rates of all fine-tuned models against the original checkpoints are below 50%, so we only compare alignment methods against one another here, rather than against the original models." This phenomenon is not specific to TGO: all fine-tuned variants under the Pick-a-Pic-based setting underperform the original FLUX checkpoint, likely due to the training-data quality mismatch.
>
> **W2: Are models trained and tested on the same reward model?**
>
> Yes. To address the concern about reward hacking, we additionally report cross-reward evaluation for models trained with HPS-based feedback and evaluated using four different reward models:
>
> | | HPSv2.1 Mean ↑ | HPSv2.1 Median ↑ | PickScore Mean ↑ | PickScore Median ↑ | ImageReward Mean ↑ | ImageReward Median ↑ | Aesthetic Mean ↑ | Aesthetic Median ↑ |
> |---|---:|---:|---:|---:|---:|---:|---:|---:|
> | SD v1.4| 0.2454 | 0.2462 | 20.8040 | 20.7784 | 0.1406 | 0.1773|5.4277|5.4293|
> | +SFT | 0.2506 | 0.2520 | 20.6996 | 20.6706 | 0.3136 | 0.3741 | 5.4475|5.4462|
> | +TGO | 0.2618 | 0.2631 | 20.9047 | 20.8774 | 0.4233 | 0.4948 | 5.4919|5.4916|
>
> The key observation is that TGO improves consistently across all four evaluators, rather than only on a single metric. Compared with the baseline model and SFT, TGO achieves better scores under all four reward models. Therefore, these cross-reward results substantially alleviate the concern that TGO's gains are due to reward hacking, and instead suggest broader improvements in generation quality.
>
> **Q1: Please discuss how Diffusion-KTO and TGO differ, and analyze sensitivity to conversion schemes or percentile thresholds.**
>
> Conceptually, Diffusion-KTO and TGO address different supervision assumptions. Diffusion-KTO is built on a Kahneman–Tversky utility-maximization view and uses binary desirable/undesirable labels. In contrast, TGO is derived from the KL-optimal decision rule for scalar feedback: it approximates the intractable instance-dependent oracle baseline with a data-driven threshold, and further leverages the score magnitude through confidence weighting. Therefore, TGO is specifically designed for native scalar ratings rather than only binarized feedback. We agree that this distinction from Diffusion-KTO should be made more explicit in related work, and we will revise that section accordingly.
>
> For the Pick-a-Pic v2 comparison, we applied the same pref-to-scalar conversion pipeline for both TGO and Diffusion-KTO to ensure a fair comparison. The key difference is how the scalar supervision is used. Diffusion-KTO directly binarizes the scalar score into labels, whereas TGO first estimates a threshold and then assigns labels relative to this threshold.
>
> We further provide an ablation over different percentile choices on Pick-a-Pic v2 training set:
>
> #### Mean Scores on Pick-a-Pic v2.
>
> | | HPSv2.1 | PickScore | ImageReward | Aesthetic | CLIPScore |
> |---|---:|---:|---:|---:|---:|
> | p=0.1 | 0.2757 | 20.93 | 0.4913 |5.66|27.44|
> | p=0.3 | 0.2841 | 21.09 | 0.6817 |5.67|27.62|
> | p=0.5 | 0.2881 | 21.13 | 0.6728 |5.63|27.72|
> | p=0.7 | 0.2690 | 20.97 | 0.3825 |5.58|27.58|
> | p=0.9 | 0.2660 | 20.89 | 0.3385 |5.53|27.82|
>
> #### Mean Scores on PartiPrompts.
>
> | | HPSv2.1 | PickScore | ImageReward | Aesthetic|CLIPScore|
> |---|---:|---:|---:|---:|---:|
> | p=0.1|0.2751|21.50|0.5695|5.55|27.14|
> | p=0.3|0.2754|21.61|0.5859|5.59|27.36|
> | p=0.5|0.2774|21.62|0.6485|5.58|27.56|
> | p=0.7|0.2616|21.46|0.4338|5.51|27.33|
> | p=0.9|0.2645|21.45|0.4643|5.43|27.39|
>
> #### Mean Scores on HPSv2.
>
> | | HPSv2.1 | PickScore | ImageReward | Aesthetic | CLIPScore |
> |---|---:|---:|---:|---:|---:|
> | p=0.1 | 0.2857 | 21.20 | 0.6720 | 5.59 | 29.68 |
> | p=0.3 | 0.2988 | 21.37 | 0.7111 | 5.63 | 29.92 |
> | p=0.5 | 0.2935 | 21.43 | 0.7514 | 5.61 | 30.17 |
> | p=0.7 | 0.2797 | 21.19 | 0.5171 | 5.59 | 30.01 |
> | p=0.9 | 0.2756 | 21.26 | 0.5308 | 5.53 | 30.23 |
>
> These results suggest that TGO is reasonably robust within a practical operating range (p=0.3–0.5), while overly high thresholds lead to degraded performance on several reward models.
>
> **Limitations section.**
>
> Our method has several limitations. First, the current formulation uses a single global threshold estimated from empirical score statistics. Although this works well in our experiments, it may be suboptimal when score distributions are strongly heterogeneous across prompts or semantic groups, in which case prompt-conditional thresholds could be more appropriate. Second, TGO does not eliminate the dependence on feedback quality: if the scalar scores are noisy, biased, or only imperfectly aligned with human preference, the induced pseudo-labels can inherit these imperfections.

---

> > ### Author Rebuttal · Reviewer_qAQF · 2026-04-03
> >
> > I appreciate the authors' effort on providing a detailed rebuttal. While I am satisfied with the responses, it's unclear why the authors didn't consider the Diffusion-KTO baseline, which is consistently the 2nd best across all datasets, for the results on cross-rewards shown above. It would have made the comparison more convincing.

---

> > > ### Author Response · Authors · 2026-04-07
> > >
> > > We thank the reviewer for this helpful suggestion. We agree that explicitly including Diffusion-KTO in the cross-reward table would make the presentation more convincing. We would like to clarify, however, that the additional cross-reward results in the rebuttal were intended to answer a narrower question in our native scalar-feedback setting, namely whether TGO trained with one reward model improves only on that same evaluator or also transfers to other evaluators. In that setting, the goal of the table was to address the reward-hacking concern for TGO itself, rather than to provide a full head-to-head benchmark of all baselines.
> > >
> > > This distinction is important because Diffusion-KTO and TGO are built for different native supervision regimes. Diffusion-KTO is formulated under the KTO utility-maximization framework with binary desirable/undesirable supervision, whereas TGO is designed for unpaired scalar feedback and further leverages score magnitude through confidence weighting. Therefore, adding Diffusion-KTO to the Setting 1 cross-reward table would not be a plug-in comparison: **it would require introducing an extra scalar-to-binary conversion on top of the native scalar-feedback setup. This introduces an additional preprocessing choice beyond original Diffusion-KTO itself**, making the interpretation of that table less clean and less directly tied to the narrower question of whether TGO is simply overfitting a single reward signal.
> > >
> > > For the direct comparison against Diffusion-KTO, we believe the more appropriate evidence is our Pick-a-Pic v2 setting. As described in Sec. 4.3 and clarified in the rebuttal, this setting was introduced precisely for controlled comparison with prior preference-alignment methods. There, we applied the same pairwise-to-scalar / pref-to-scalar conversion pipeline so that TGO and Diffusion-KTO operate on matched supervision signals, avoiding confounds from different preprocessing. **In this sense, Setting 2 serves as the matched comparison to existing baselines, whereas Setting 1 serves to validate TGO in the true scalar-feedback regime that motivates the paper.**
> > >
> > > Relatedly, the reward-hacking concern is also different across the two settings. In Setting 1, the training signal is indeed derived from reward-model scores, which is exactly why we added cross-reward evaluation. In Setting 2, by contrast, the training signal is derived from human preference annotations in Pick-a-Pic v2, aggregated into scalar win counts and then thresholded, rather than from the same reward model used for evaluation. Thus, the usual same-reward-model train/test concern is not the primary issue in that setting. From this perspective, the existing Pick-a-Pic comparison already provides the more appropriate evidence for TGO’s effectiveness relative to Diffusion-KTO under a matched protocol.
> > >
> > > That said, we agree that explicitly adding Diffusion-KTO to the cross-reward table would further strengthen the presentation. Our intention here was to address the narrower reward-hacking question in Setting 1, while the direct comparison to Diffusion-KTO is provided in Setting 2 under a matched protocol.

---

### Decision · Program_Chairs · 2026-04-30

**Decision:**

Accept (regular)

**Comment:**

The reviewers agree that this paper addresses a practically relevant problem of aligning visual generative models from scalar feedback, and they find the submission clear and technically sound. Strengths highlighted across reviews include the formulation of threshold-guided optimization as an effective approach for learning from unpaired scalar scores and an interesting connection to the KL-regularized perspective.

The main concerns were the use of a global threshold, the scope of the theoretical guarantees, and questions about novelty and reward-specific effects. Based on my reading of the rebuttal and discussion, I find that the authors provided sufficient additional evidence, including threshold sensitivity analysis, conditional threshold variants, and comparison to an additional baseline. While some comments about conceptual simplicity remain, there is agreement that the problem is meaningful and the numerical results are strong.